



# A computationally efficient engineering aerodynamic model for non-planar wind turbine rotors

Ang Li[1], Mac Gaunaa[1], Georg Raimund Pirrung[1], and Sergio González Horcas[1]

[1]Department of Wind Energy, Technical University of Denmark, Frederiksborgvej 399, DK-4000 Roskilde, Denmark

**Correspondence:** Ang Li (angl@dtu.dk)

**Abstract.** In the present work, a computationally efficient engineering model for the aerodynamic load calculation of non-planar wind turbine rotors is proposed. The method is based on the vortex cylinder model, and can be used in two ways: either as a correction to the currently widely used blade element momentum (BEM) method, or used as the main model, replacing the BEM method in the engineering modelling complex. The proposed method needs the same order of computational effort as the ordinary BEM method, which makes it ideal for time-domain aero-servo-elastic simulations. The results from the proposed method are compared with results from two higher-fidelity aerodynamic models: a lifting-line method and a Navier-Stokes solver. For planar rotors, the aerodynamic loads are identical to the current BEM model when the drag force is excluded during the calculation of the induced velocities. For non-planar rotors, the influence of the blade out-of-plane shape, measured by the difference of the load between the non-planar rotor and the planar rotor, is in very good agreement with higher-fidelity models. Meanwhile, the existing BEM methods, even with a correction of radial induction included, show relatively large deviations from the higher-fidelity method results.

## 1 Introduction

The blade element momentum (BEM) method has long been dominant in the low-fidelity aerodynamic modelling of horizontal-axis wind turbines. Until now, it is the main working horse for wind turbine aero-servo-elastic simulations and is widely used in the wind turbine design and optimization framework. There are many explicit and implicit assumptions in the BEM method. The BEM method explicitly assumes that uniform inflow is applied to the rotor that is operating at a high tip-speed ratio and the stream tubes are independent of each other. The model also implicitly assumes a planar rotor with straight blades and using quasi-steady aerodynamics. There has been extensive work on the modifications and corrections to the BEM method, such as dynamic stall model (Leishman and Nguyen, 1990; Hansen et al., 2004; Larsen et al., 2007), dynamic inflow model (Schepers and Snel, 1995; Yu et al., 2019), polar-grid based unsteady BEM (Madsen et al., 2020a), modelling of turbulent inflow (Mann, 1994), high-thrust correction (Spera, 1994; Madsen et al., 2020a; Burton et al., 2021) and corrections for operation in yawed conditions (Leishman, 2005).





The results from the BEM method generally show surprisingly good agreement with higher-fidelity models, at least on the integral level. However, due to the progress of wind turbine technology, modern multi-megawatt designs are generally more flexible than the stiff machines of the 1980s. It implies that modern wind turbine blades typically have more prebend, larger cone angle and larger deformations. The influence of blade out-of-plane shapes on the aerodynamics is then more pronounced

and can not simply be neglected. Some new developments have even more pronounced out-of-plane shapes. For example, a downwind wind turbine designed for low-wind conditions could have large cone and prebend and possibly dramatic out-of-plane deformations (Madsen et al., 2020b). In addition, some wind turbines are equipped with winglets to reduce the drag force and also the noise. Modern wind turbines are generally designed using mainly the BEM-based codes. Higher-fidelity tools such as lifting-line method (LL) or fully-resolved Navier-Stokes solvers (often referred to as Computational Fluid Dynamics, CFD),

are mostly used for comparison or for very specific load cases due to the high computational effort. However, when the blades have large out-of-plane shapes due to prebend, deformation or cone, the results from these BEM codes will have relatively large differences compared to the results from higher-fidelity tools. This is because the influence of blade out-of-plane shapes is not correctly captured by these BEM-based codes. As a result, a design from an optimization tool using a BEM-based aerodynamics module could be far from the actual optimal solution. There may even exist aeroelastic instabilities that are not

correctly captured by the BEM-based tools.

On the other hand, rotor-resolved CFD and lifting-line method are computationally too expensive for extensive use in current and near-future design optimization process. Navier-Stokes solvers with fully resolved rotor geometry have no difficulties predicting non-planar rotor effects. But for the lifting-line method, care should be taken on the influence of curved bound vortex on itself (Li et al., 2020), the correct directions of applying the lift and drag force and also the possible non-circulatory

lift during the aerodynamic load calculation, to correctly predict the effects. Therefore, a low-fidelity model that could capture the most important features of the aerodynamics of non-planar rotors, while maintaining approximately the same level of computational effort as the current BEM methods, would be of great value to both the scientific and the commercial wind turbine communities. Both for the design optimization as well as for the aeroelastic simulations.

In order to correctly account for the out-of-plane shapes of the wind turbine blades in the low-fidelity model, the physics be-

hind the problem should be analyzed and then the most important aspects should be proactively modelled while less important features can be neglected. In the present work, the force on the non-planar rotor is firstly analyzed in a physically consistent way using the Kutta-Joukowski theorem. The conclusion from the analysis is that the streamwise-shifted starting position of the trailed vorticity, due to the non-planar bound vortex surface swept by the blades, will influence both axial and radial induction and both have direct influences on the aerodynamic loads. Therefore, we consider the vortex cylinder model (Branlard and

Gaunaa, 2015a) has the potential to capture these most important features in the aerodynamics of the non-planar rotors. There has been previous work by Crawford (2006) using a vortex cylinder model for the aerodynamic calculation of coned rotors. In that work, the same idea of using the axial and radial induction from the vortex cylinder model is proposed, and the closed equations for the model are given in the framework of momentum theory. Some comparisons with actuator disc results for uniform-loaded cases are shown for the planar rotor. And for the coned rotors, the axial induction as well as aerodynamic loads

are compared with the results from actuator disc simulations performed by Mikkelsen (2004). The good agreement shows





the potential of the vortex cylinder model for the non-planar rotor. However, the work is limited to the momentum theory framework while the equivalence of the vortex cylinder model and the momentum theory for planar rotors is not highlighted. Furthermore, other important features of the vortex cylinder model for non-planar rotors, such as the similarity to the planar rotors or the impact of unsteady airfoil aerodynamics on the steady-state results, are not described. Nevertheless, the pioneering

work by Crawford (2006) inspires the authors and is a good starting point for the current study, which also builds on previous efforts on superposition of vortex cylinders (Branlard and Gaunaa, 2015a).

In the present work, a detailed analysis of the vortex cylinder model for non-planar rotors will be performed. A method based on the vortex cylinder model for the aerodynamic load calculation of such non-planar rotors is then proposed. The description of the implementation of the proposed method is in the framework of the HAWC2 code (Larsen and Hansen, 2007). Some

details of the implementation may be different compared to other BEM-based aeroelastic codes. In the present work, only the out-of-plane shape of the blade is considered, which means the blade is assumed to have no in-plane sweep. The engineering aerodynamic model for the blades with only in-plane shapes is described by Li et al. (2021). The structure of the work is as follows: the Kutta-Joukowski analysis of the non-planar rotor is performed in Sect. 2. Then, the vortex cylinder model as well as its relationship with the momentum theory are briefly introduced in Sect. 3. Some important aspects for the implementation

of the BEM method and the proposed method for non-planar rotors are described in Sect. 4. The coupling of the blade element theory with the vortex cylinder model, including details on the tip-loss correction and a summary of the algorithm, is described in Sect. 5. The low-fidelity and higher-fidelity aerodynamic models for comparisons are described in Sect. 6. The setup of the test cases is described and the results from the low-fidelity models are compared with the results from higher-fidelity models in Sect. 7. Finally, the conclusions are drawn and the future work is summarized in Sect. 8.

## 2  Kutta-Joukowski analysis

For the planar rotor with straight blades, the Kutta-Joukowski analysis was previously used to derive the similarity between the superposition of the vortex cylinders and the BEM method by Branlard and Gaunaa (2015a). In this section, the influence of the blade out-of-plane shapes on the aerodynamics is investigated using the Kutta-Joukowski theorem (Okulov et al., 2015). The blade is assumed only possible to have out-of-plane shapes (dihedral or cone) but no in-plane shapes (blade sweep) in the

following analysis.

The coordinate system is defined as follows and is illustrated in Fig. 1. The $x-$axis is the axial direction, positive in the incoming wind direction. The rotation vector of the rotor is in the positive $x-$direction. The $y-$axis is the 'radial' direction, which is positive in the direction of increasing radius of the blade. The $z-$axis is normal to both $x-$axis and $y-$axis, and its direction is defined so that a right-handed system is found. The $z-$axis is defined as the tangential direction. The airfoils are

aligned perpendicular to the main-axis of the half-chord line.

The local dihedral angle is defined to be positive when the blade is tilting upwind, and can be calculated using the blade main-axis geometry.

$$\kappa_i = -\arctan \frac{\mathrm{d}x}{\mathrm{d}y}(y_i) \tag{1}$$



The analysis is applied to a non-planar rotor with given blade bound circulation and induced velocity at each blade section. For section $i$ of a blade with radius of $r_i$, the bound circulation strength is $\Gamma_i^B$, the local dihedral angle is $\kappa_i$. The axial, tangential and radial induced velocity are $u_{a,i}$, $u_{t,i}$ and $u_{r,i}$.

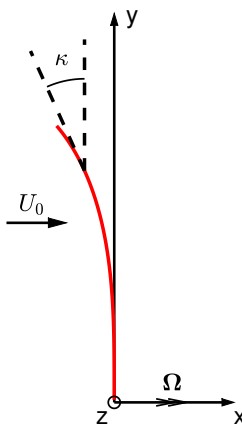

**Figure 1.** The definition of the coordinate system of the wind turbine with only out-of-plane shapes. The $x-$axis is the axial direction, positive in the incoming wind direction. The $y-$axis is the 'radial' direction, which is positive in the direction of increasing radius of the blade. The $z-$axis is normal to both $x-$axis and $y-$axis, and its direction is defined so that a right-handed system is found. The rotation vector $\mathbf{\Omega}$ is aligned with the $x-$axis.

The relative velocity experienced by the blade section is $\boldsymbol{V_{rel,i}}$. The bound circulation of a blade section $\boldsymbol{\Gamma_i^B}$ is tangent to the

5   local blade section.

$$\boldsymbol{V_{rel,i}} = \begin{pmatrix} U_0 + u_{a,i} \\ u_{r,i} \\ -\Omega r_i + u_{t,i} \end{pmatrix}, \quad \boldsymbol{\Gamma_i^B} = \Gamma_i^B \begin{pmatrix} -\sin\kappa_i \\ \cos\kappa_i \\ 0 \end{pmatrix} \tag{2}$$

With the Kutta-Joukowski theorem in three-dimensional vector form, the lift force on the blade is obtained.

$$\boldsymbol{f_i} = \rho \boldsymbol{V_{rel,i}} \times \boldsymbol{\Gamma_i^B} = \rho \Gamma_i^B \begin{pmatrix} (\Omega r_i - u_{t,i})\cos\kappa_i \\ (\Omega r_i - u_{t,i})\sin\kappa_i \\ (U_0 + u_{a,i})\cos\kappa_i + u_{r,i}\sin\kappa_i \end{pmatrix} \tag{3}$$

In Eq. (3), the force $\boldsymbol{f_i}$ is corresponding to lifting force per unit curved blade length. The lifting force per unit radius,

10   corresponding to what is used in momentum theory analysis, is $\boldsymbol{f_i^*} = \boldsymbol{f_i}\frac{ds}{dr}$, as also shown by Madsen et al. (2020a). The $\frac{ds}{dr}$ term is representing the ratio of the local change of curved blade length and local change of radius. For blades with only



out-of-plane shapes, it is equal to: $1/\cos\kappa_i$.

$$\boldsymbol{f_i^*} = \boldsymbol{f_i}\frac{\mathrm{d}s}{\mathrm{d}r} = \rho\Gamma_i^{\mathrm{B}}\begin{pmatrix} \Omega r_i - u_{t,i} \\ (\Omega r_i - u_{t,i})\tan\kappa_i \\ (U_0 + u_{a,i}) + u_{r,i}\tan\kappa_i \end{pmatrix} \tag{4}$$

The force $\boldsymbol{f_i^*}$ is divided into two parts: a part with and a part without the direct contribution of the local dihedral angle $\kappa_i$.

$$\boldsymbol{f_i^*} = \rho\Gamma_i^{\mathrm{B}}\begin{pmatrix} \Omega r_i - u_{t,i} \\ 0 \\ U_0 + u_{a,i} \end{pmatrix} + \rho\Gamma_i^{\mathrm{B}}\begin{pmatrix} 0 \\ \Omega r_i - u_{t,i} \\ u_{r,i} \end{pmatrix}\tan\kappa_i \tag{5}$$

For a non-planar rotor with upwind direction dihedral ($\kappa_i > 0$), such as prebend or upwind cone, some conclusions can be obtained according to Eq. (5). Comparing to the corresponding planar rotor (as shown later on in Fig. 3), the non-planar rotor will have outboard radial force. Furthermore, there will be additional tangential driving force due to the radial induction, which for the wind turbine case is positive. So, to get the correct tangential load distribution and consequently the aerodynamic power, it is not only necessary to correctly model the influence of the non-planarity of the rotor on the axial and tangential induced

velocity, but also on the radial induced velocity. However, the radial induced velocity is not available from the momentum theory.

There are two tracks to modify the BEM method to model the non-planar effects. The first track is based on the previous work on the development of the radial induction correction for the application in the BEM method, derived based on analytical 2-D actuator disc/strip model combined with an engineering fit of the numerical actuator disc simulations (Madsen, 1997;

Madsen et al., 2010):

$$u_r^{\mathrm{Madsen}}(r) = \frac{U_0}{2.24}\frac{C_{T,av}(r)}{4\pi}\ln\left[\frac{0.04^2 + (\frac{r}{R_{\mathrm{tot}}} + 1)^2}{0.04^2 + (\frac{r}{R_{\mathrm{tot}}} - 1)^2}\right] \tag{6}$$

where $C_{T,av}(r)$ is the averaged thrust coefficient as function of the radial position and is defined as:

$$C_{T,av}(r) = \frac{\int_0^r C_T(r^*)2\pi r^* \,\mathrm{d}r^*}{\pi r^2} \tag{7}$$

However, the radial induction as well as the axial induction are corresponding to planar rotors.

Apart from the momentum theory, which effectively only applies to a planar rotor disc, it is possible to calculate the induced velocity, including the radial component, from analytical equations at each blade section with the superposition of vortex cylinders. So, the second track is based on the vortex cylinder model where the assumption of the planar rotor in the previous work by Branlard and Gaunaa (2015a) is relaxed. This approach inherently includes the effect of axial displacement of the cylindrical wake of the non-planar rotor in a physically consistent manner. The model even has the potential to completely

replace the momentum theory in the BEM method and will be described in the following sections. In the present work, both methods will be tested numerically in Sect. 7.





## 3 Vortex cylinder model

The vortex cylinder model is a simplified representation of the vortex system of a horizontal-axis wind turbine rotor. The model consists of superposition of bound vortex discs, non-expanding vortex cylinders with both tangential and longitudinal vorticity and root vortices (Branlard and Gaunaa, 2015a). It could be considered as the special condition of the Joukowski rotor model for the limiting case of infinite number of blades. It has been shown by Branlard and Gaunaa (2015a) that for a planar rotor, the induced velocities from the version of the vortex cylinder model where the wake rotation effect is neglected, is identical to the induced velocities obtained from the momentum theory when the drag force component is not included in the force balancing from which the induced velocities are calculated. It was also argued by Branlard (2017) that the correct way of implementing the blade element momentum (BEM) method should exclude the drag during the calculation of the induced velocities, and include the drag in the aerodynamic load calculation afterwards.

One advantage of the vortex cylinder model over the momentum theory is that the induced velocity at any arbitrary point in the flow field is known. In contrast, from the momentum theory, only the axial and tangential velocity at the rotor disc and at infinitely far up- and down-stream of the rotor plane are known. This advantage has been used in the application of the vortex cylinder model in the calculation of the induction-zone of a wind turbine (Branlard and Meyer Forsting, 2015) and the wind farm blockage effects (Branlard and Meyer Forsting, 2020). Another advantage of the vortex cylinder model over the momentum theory is that it does not require the assumptions of a planar rotor and the flow with constant speed being perpendicular to it. Other applications of the vortex cylinder model include modelling of wind turbines in yaw (Branlard and Gaunaa, 2016) and modelling of the dynamic inflow effects (Yu et al., 2019). The results from all the aforementioned applications compare well with higher-fidelity tools, indicating that the main mechanisms are captured using this framework.

When applying the vortex cylinder method to a non-planar rotor, the starting position of the cylindrical vortex sheets follows the curved bound vortex surface and will be displaced upstream or downstream compared to the case of a planar rotor. The induced velocity on the non-planar rotor surface will therefore be different from the induced velocity of a planar rotor. This effect can be modelled by the superposition of the vortex cylinders according to the curved bound vortex surface swept by the blades that have out-of-plane shapes. However, the possibility of using the vortex cylinder model for the non-planar rotor is not well recognized and is thus not widely utilized. The work of Crawford (2006) is on this topic, but the system closure for the non-planar rotor is different compared to the current study. In the current implementation, the system closure is determined in the far-wake, thus assuming the same method of system closure as for the non-planar rotors. Then, the equations of the inductions of the non-planar rotor are in concise forms following this assumption. With the current system closure, which is an important assumption in this work, there are clear physical connections between the vortex cylinder model of a non-planar rotor and a planar rotor, and subsequently the connection to the momentum theory. In the following content of this work, zero yaw error, no rotor tilt and uniform inflow are assumed. The vortex cylinder is then a right cylinder as opposed to an oblique cylinder used in yawed flow analysis (Branlard and Gaunaa, 2016).





## 3.1 The right vortex cylinder

The equations of the inductions of a right vortex cylinder have been derived in detail by Branlard and Gaunaa (2015b). The most important equations and conclusions are summarized in this section. A cylindrical vortex sheet can be decomposed into tangential and longitudinal vorticity components. The strength of the tangential vorticity on the vortex cylinder is the ratio of

the total vorticity strength to the helical pitch $h$. The total vorticity strength of the vortex cylinder $\Delta\Gamma_{\text{tot}}$ is equal to the trailed vorticity strength of all blades.

$$\gamma_t = -\frac{\Delta\Gamma_{\text{tot}}}{h} \tag{8}$$

The tangential vorticity contributes to both axial and radial induced velocities. For the vortex cylinder with radius $R$ at axial position $x = 0$, the axial and radial induced velocity at the calculation point with radius of $r$ and axial position of $x$ are shown

to be in the form of complete elliptic integrals (Branlard and Gaunaa, 2015b).

$$u_a(r,x) = \frac{\gamma_t}{2}\left[\frac{R-r+|R-r|}{2|R-r|} + \frac{xk(r,x)}{2\pi\sqrt{rR}}\left(K\left(k^2(r,x)\right) + \frac{R-r}{R+r}\Pi\left(k^2(r,0),k^2(r,x)\right)\right)\right] \tag{9}$$

$$u_r(r,x) = -\frac{\gamma_t}{2\pi}\sqrt{\frac{R}{r}}\left[\frac{2-k^2(r,x)}{k(r,x)}K\left(k^2(r,x)\right) - \frac{2}{k(r,x)}E\left(k^2(r,x)\right)\right] \tag{10}$$

where

$$k^2(r,x) = \frac{4rR}{(R+r)^2 + x^2} \tag{11}$$

and $K(m)$, $E(m)$ and $\Pi(n,m)$ are the complete elliptic integral of the first, second and third kind.

The other components of the vortex cylinder, which are the bound vortex disc, the longitudinal vorticity and the root vortex line, only have contribution to the tangential velocity. The tangential induced velocity of the entire flow field is derived by Branlard and Gaunaa (2015b) as follows:

$$u_t(r,x) = \begin{cases} -\frac{\Delta\Gamma_{\text{tot}}}{4\pi r} & (r < R, x = 0) \text{ or } (r = R, x > 0) \\ -\frac{\Delta\Gamma_{\text{tot}}}{2\pi r} & (r < R, x > 0) \\ 0 & \text{otherwise (outside the vortex cylinder)} \end{cases} \tag{12}$$

## 3.2 Superposition of vortex cylinders for planar rotors

Consider the superposition of the Joukowski rotor model to achieve radially varying bound circulation. There will be helical trailed vorticities emanated along each blade with the strength equal to the derivative of the bound circulation strength with respect to the radius. In Joukowski's rotor model, it is assumed that the radial distribution of the bound circulation of all blades are the same. Then consider the corresponding vortex cylinder model that is the limiting case of Joukowski's rotor model, where

the number of blades tends to infinity. It is consisted of a superposition of cylindrical vortex sheets with both tangential and



longitudinal vorticity. Details of the superposition of vortex cylinders have been described by Branlard and Gaunaa (2015a). The most important aspects are summarized in this section.

With the superposition of the vortex cylinders, the bound circulation is assumed to be piecewise constant along the blade. The blade is discretized radially into $n$ sections and there will be one calculation point for each section. Consequently, there

will be $(n+1)$ trailing points corresponding to $(n+1)$ vortex cylinders. For the inner-most part of the rotor, which is usually the rotor hub and is defined from the center of rotation to the beginning of the first blade section, the bound circulation strength is zero since there are no blades. For the ease of notation and calculation, two ghost sections with the index of 0 and $n+1$ with zero circulation strength are introduced. For the system, the number of unknown variables is $n$, which is equal to the number of sections. A sketch of the superposition of the cylindrical vortex system is shown in Fig. 2.

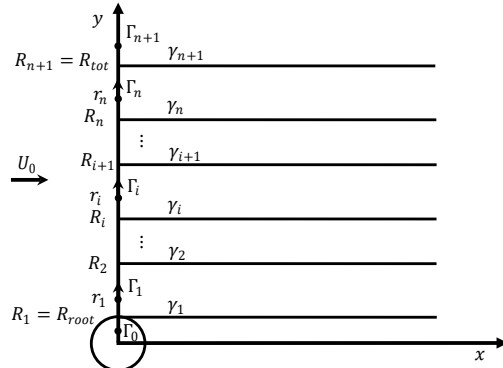

**Figure 2.** Sketch of the superposition of the cylindrical vortex system. The blade is extending from $R_{\text{root}}$ to $R_{\text{tot}}$ and is discretized into $n$ sections. For section $i$, the radius of this calculation point is $r_i$ and the two neighbouring vortex cylinders are with the radius of $R_i$ and $R_{i+1}$, and with the tangential vorticity strength of $\gamma_{t,i}$ and $\gamma_{t,i+1}$. Two ghost sections with the index of 0 and $n+1$ are introduced. These two ghost sections are defined to have zero bound circulation strength.

The closure of the system determines the tangential vorticity strength of each vortex cylinder. The closure of the system is determined at the far-wake, the cylindrical vortex sheet is assumed to convect at a constant speed equal to the mean of the two far-wake velocities surrounding the vortex sheet (Branlard and Gaunaa, 2015a). It can be shown that the system closure can be performed in the form of helical pitch $h(r)$ or the annulus axial induction factor $a(r)$ at the rotor disc, and the two formulations are equivalent to each other. The system closure in the form of the annulus axial induction factor will be used in this work and

will be briefly described.

With the system closure, the axial induction factor of section $i$ is calculated as follows:

$$a_i = -\frac{u_{a,i}}{U_0} = \frac{1}{2}\left[1 - \sqrt{1 - C_{T,\text{eff},i}}\right] \tag{13}$$

The effective thrust coefficient $C_{T,\text{eff},i}$ is equal to the thrust coefficient from the Kutta-Joukowski analysis $C_{T,\text{KJ},i}$ minus the contribution of wake rotation $C_{T,\text{rot},i}$.

$$C_{T,\text{eff},i} = C_{T,\text{KJ},i} - C_{T,\text{rot},i} \tag{14}$$



where

$$C_{T,\text{KJ},i} = \frac{N_B f_{i,x}^* \, \mathrm{d}r}{\frac{1}{2}\rho U_0^2 2\pi r_i \, \mathrm{d}r} = k_{s,i}\left(1 + \frac{k_{s,i}}{4\lambda_{r_i}^2}\right) = k_{s,i}\left(1 + a_i'\right) \tag{15}$$

$$C_{T,\text{rot},i} = \sum_{j \geq i+1} \left(\frac{k_{s,j}}{2}\right)^2 \left(\frac{1}{\lambda_{R_j}^2} - \frac{1}{\lambda_{R_{j+1}}^2}\right) \tag{16}$$

The wake rotation effect increases toward the rotor rotational axis, and decreases when the tip-speed ratio increases. For typical modern wind turbine designs the effect of this term is rather small and may be neglected.

In Eqs. (15) and (16), $k_{s,i}$ is the total non-dimensional bound circulation of section $i$ and $\lambda_r$ is the local speed ratio for the position with radius $r$. The tangential induction factor $a_i'$ is defined as follows and can be calculated according to Eq. (12) with the condition of $x = 0$.

$$k_{s,i} = \frac{\Omega \Gamma_i}{\pi U_0^2} \tag{17}$$

$$\lambda_r = \frac{\Omega r}{U_0} \tag{18}$$

$$a_i' = \frac{-u_{t,i}}{\Omega r_i} = \frac{\sum_{j=i}^{n} \frac{\Gamma_j - \Gamma_{j+1}}{4\pi r_i}}{\Omega r_i} = \frac{k_{s,i}}{4\lambda_{r_i}^2} \tag{19}$$

Considering Eqs. (15) and (16), when the wake rotation effect is included, the aerodynamic loading of a section is dependent on all sections that are further outboard comparing to it. The system could be solved from outboard to inboard using Eq. (13), and the system closure is completed when the annulus axial induction factor of all blade sections are calculated. Then, the tangential vorticity of the vortex cylinder that is just inside section $i$ is obtained from the annulus axial induction factor of this section and the neighbouring section inside. The equation can be derived from Eq. (9) using the condition of $x = 0$.

$$\gamma_{t,i} = 2U_0(a_i - a_{i-1}) \tag{20}$$

### 3.3 High-thrust correction

In the vortex cylinder model, the relationship between the axial induction factor and the effective thrust coefficient is in the same form as in the momentum theory: $C_{T,\text{eff},i} = 4a_i(1 - a_i)$, by inversing Eq. (13). However, when the thrust coefficient ($C_T$) is high, especially when $C_T > 1$, the momentum theory breaks down and the vortex cylinder model will give unphysical results. Then, corrections should be made for these high-thrust conditions. Different high-thrust corrections are available, such as the linear extrapolation by Spera (1994) and the polynomial function of $a$ and $C_T$ by Madsen et al. (2020a). To be consistent with the BEM module implemented in the HAWC2 code (Larsen and Hansen, 2007), the polynomial function of $a$ and $C_T$ by Madsen et al. (2020a) is chosen. Then, in the system closure, the equation of axial induction factor from the thrust coefficient in Eq. (13) should be replaced by:

$$a_i = f_{a-C_T}^{\text{Madsen}}(C_{T,i}) = k_3 C_{T,i}^3 + k_2 C_{T,i}^2 + k_1 C_{T,i} \tag{21}$$

where the coefficients $k_1 \ldots k_3$ are defined: $k_1 = 0.2460$, $k_2 = 0.0586$ and $k_3 = 0.0883$.



### 3.4 System closure of non-planar rotors

The system closure of the vortex cylinder model for planar rotors has been described in Sect. 3.2. It is assumed that each vortex cylinder convects with a constant velocity that is determined in the far-wake. With this method of system closure, the relationship between the vortex cylinder model and the momentum theory is revealed by Branlard and Gaunaa (2015b) and the results are generally in good agreement with higher-fidelity models. The reason is probably that the error introduced when assuming the convective velocity is constant balances the error introduced when assuming the non-expanding wake as shown for the uniformly loaded disc (Øye, 1990; Madsen et al., 2007). In the present work, we assume the same method of system closure as for the planar rotor case: the closure is determined at the far-wake, and can be used for non-planar rotors with moderate out-of-plane shapes. With this assumption, the equations are in concise forms and can clearly show the connections between the model for a non-planar rotor and a planar rotor. However, this assumption does not necessarily hold for extreme cases. To investigate this, a numerical test of blades with relatively large prebend and cone angle will be shown in Sect. 7.

For illustration, we show the superposition of the vortex cylinders for a non-planar rotor and the corresponding planar rotor with the same radial discretization in Fig. 3. For each section, the corresponding planar rotor has the same total bound circulation (of all blades) $\Gamma_i^{\mathrm{pl}}$ as that of the non-planar rotor $\Gamma_i^{\mathrm{np}}$.

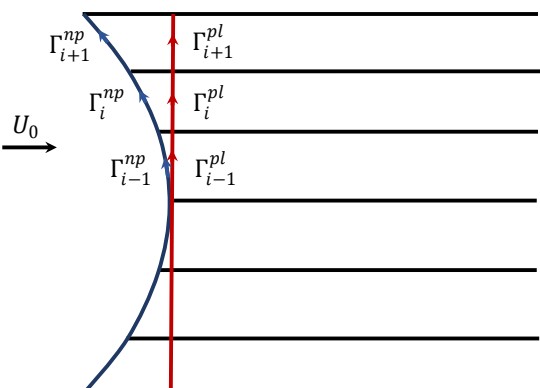

**Figure 3.** Side-view of the vortex system of the non-planar rotor and the corresponding planar rotor. The two vortex systems are with the same radial discretization and radial distribution of bound vorticity. The tangential and longitudinal trailed vorticity strengths of the two systems will be identical.

### 3.4.1 Similarity of thrust coefficient

The first similarity is the calculation of the thrust coefficient of the non-planar rotor and the corresponding planar rotor. For the non-planar rotor, the Kutta-Joukowski thrust coefficient of section $i$ is calculated from the $x-$component of the force in





Eq. (5), obtained using the Kutta-Joukowski analysis.

$$C_{T,\text{KJ}}^{\text{np}} = \frac{\Omega \Gamma_i^{\text{np}}}{\pi U_0^2}(1 + a'^{\text{np}}) \tag{22}$$

For the planar rotor, the thrust coefficient is also obtained using Eq. (5):

$$C_{T,\text{KJ}}^{\text{pl}} = \frac{\Omega \Gamma_i^{\text{pl}}}{\pi U_0^2}(1 + a'^{\text{pl}}) \tag{23}$$

Since $\Gamma_i^{\text{np}} = \Gamma_i^{\text{pl}}$, and when assuming the tangential induction factor $a'$ of the non-planar rotor and the corresponding planar rotor are the same, which will be proven analytically in Sect. 3.4.3, then the Kutta-Joukowski thrust coefficient of the non-planar rotor and the planar rotor are identical. In addition, the contribution of the wake rotation to the thrust coefficient will also be identical. So, with the given bound circulation distribution of the non-planar rotor, the thrust coefficient distribution can be directly calculated as if the rotor is planar. This is true no matter the wake rotation effect is included or excluded.

### 3.4.2 Similarity of tangential and longitudinal vorticity in the wake

The second similarity is that the two vortex wake systems are having the same tangential and longitudinal vorticity strength distribution. For the two rotors with the same radial distribution of bound circulation, it can be easily shown that the trailed vorticity strengths between each section are the same.

$$\Gamma_i^{\text{np}} - \Gamma_{i+1}^{\text{np}} = \Gamma_i^{\text{pl}} - \Gamma_{i+1}^{\text{pl}} \tag{24}$$

According to the assumption, the closure of the superposition of the vortex cylinders is determined at the far-wake (infinitely far downstream). Therefore, there is no influence of the changed starting position of the vortex cylinders. As a result, both the tangential and longitudinal vorticity of the non-planar rotor wake is the same as that of the corresponding planar rotor that has the same bound circulation distribution.

According to the description in Sect 3.4.1, the annulus axial induction of the corresponding planar rotor can be calculated from the thrust coefficient of the non-planar rotor since the thrust coefficient of the two rotors are identical. Then, since the non-planar rotor and the corresponding planar rotor have the same tangential vorticity strength, the tangential vorticity strength of the non-planar rotor can be calculated from the annulus axial induction factor of the corresponding planar rotor using Eq. (20). This means with given bound circulation distribution, the tangential vorticity of the non-planar rotor can also be calculated as if the rotor is planar. The same argument can also be made for the longitudinal vorticity in the wake.

### 3.4.3 Similarity of tangential induction

It is assumed that the radial distribution of the tangential induction factor of the non-planar rotor is the same as the corresponding planar rotor in Sect. 3.4.1, this will be analytically proven in this section. Firstly, consider the superposition of the planar vortex cylinders consisting of bound vortex discs as well as tangential and longitudinal trailed vorticities, as shown in Fig. 3. The total strength of the root vortex is zero. Since the tangential vorticities have no contribution to the tangential induction,



only the bound vortex discs and the longitudinal vorticity are considered here. For a better illustration, the side view of a part of the axisymmetric vortex system that was illustrated in Fig. 3 is shown in Fig. 4.

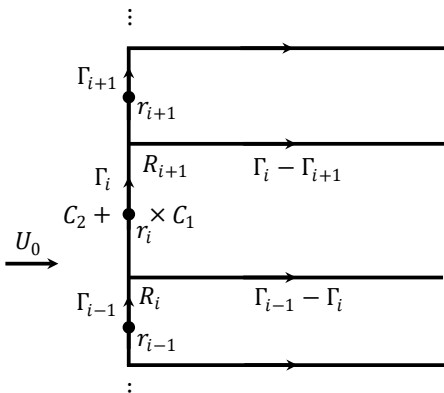

**Figure 4.** Side-view of a part of the axisymmetric vortex system of the planar rotor. The bound vorticity and trailed vorticity are highlighted. The two points of interest are marked with $\times$ and $+$, which are just inside and outside the section $i$. The two circular contours of $C_1$ and $C_2$ pass through the two points of interest respectively.

Consider the point $\times$ in Fig. 4 with radial position of $r_i$ that is between $R_i$ and $R_{i+1}$, and is just inside the vortex cylinder $(x = 0^+)$. The circular contour line $C_1$ is perpendicular to the flow, passes point $\times$ with radius $r_i$ and centered on the line of
5    $r = 0$. With the axisymmetry of the flow, the tangential velocity has the same value all along the circular contour.

Recall the definition of circulation:

$$\Gamma = \oint_C \boldsymbol{V} \cdot \mathrm{d}\boldsymbol{l} \tag{25}$$

The relationship between the velocity along the contour line $C_1$, which is the tangential velocity at $r_i$ and the net circulation through the contour $C_1$, is obtained using the definition of circulation in Eq. (25).

$$10 \quad \sum_{j=1}^{i} \left( \Gamma_{j-1} - \Gamma_j \right) = -\Gamma_i = u_t(r_i, 0^+) \, 2\pi r_i \tag{26}$$

So, the tangential velocity at point $\times$ that is just inside the vortex cylinder is:

$$u_t(r_i, 0^+) = -\frac{\Gamma_i}{2\pi r_i} \quad (R_i < r_i < R_{i+1}) \tag{27}$$

For the point $+$ in Fig. 4 with radial position of $r_i$ and is just outside the vortex cylinder $(x = 0^-)$, consider the circular contour line $C_2$ with radius $r_i$ that is perpendicular to the flow, passing point $+$ and is centered on the line of $r = 0$. Similarly,
15   use the definition of circulation in Eq. (25), the tangential velocity at point $+$ is zero since there is no net circulation passing through the contour $C_2$.

$$u_t(r_i, 0^-) = 0 \tag{28}$$





There is a jump of the tangential velocity when the flow passes through the bound vorticity disc. The tangential velocity at the disc should be the mean value of the tangential velocities at the two sides of the disc.

$$u_t(r_i,0) = \frac{1}{2}\left[u_t(r_i,0^-) + u_t(r_i,0^+)\right] = -\frac{\Gamma_i}{4\pi r_i} \quad (R_i < r_i < R_{i+1}) \tag{29}$$

The same result was obtained for the planar rotor case by Branlard and Gaunaa (2015b) by evaluation of the contribution of each component of the cylindrical vortex system to the tangential velocity. As for the planar rotor, the side view of a part of the vortex system of the non-planar rotor that was illustrated in Fig. 3 is shown in Fig. 5.

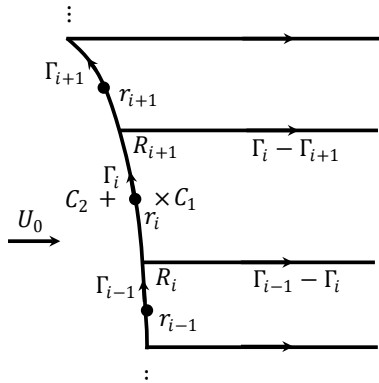

**Figure 5.** Side-view of a part of the axisymmetric vortex system of the non-planar rotor. The bound vorticity and trailed vorticity are highlighted. The two points of interest are marked with × and +, which are just inside and outside the section $i$. The two circular contours of $C_1$ and $C_2$ pass through the two points of interest.

Similar as for the planar rotor case, consider the point × in Fig. 5 with radial position of $r_i$ that is between $R_i$ and $R_{i+1}$, and is just inside the vortex cylinder. With the axisymmetry of the flow and the definition of the circulation in Eq. (25), the tangential velocity is derived to be identical to the planar rotor case in Eq. (27). It is important to point out that the net circulation through the contour ($C_1$ or $C_2$) is not influenced by the path of the circulation on either side of the contour.

For the point + that is just outside the vortex cylinder, the tangential velocity is derived to be zero and is identical to the planar rotor case in Eq. (28). Then, with the same argument as for the planar rotor, the tangential velocity at the curved bound vortex surface of the non-planar rotor should be the mean value of the tangential velocities at the two sides and is in identical form as that for the planar rotor case in Eq. (29).

## 3.5 Relationship between vortex cylinder model and momentum theory

The relationship between the vortex cylinder model and the momentum theory will be described separately for the planar rotor and the non-planar rotor.





### 3.5.1 Planar rotor

For a planar rotor at a high tip-speed ratio, when excluding the contribution of drag to the momentum balancing for determining the induced velocities, the converged results from the momentum theory are equal to those from the vortex cylinder model (Branlard and Gaunaa, 2015a). As the tip-speed ratio decreases, results from vortex cylinder model and basic 2-D momentum

theory start differing, especially toward the rotor axis. As also shown in (Branlard and Gaunaa, 2015a), this difference stems from the pressure drop caused by centrifugal forces due to wake rotation, which is not included in the classic 2-D momentum framework that the BEM method is built on. The contribution of wake rotation to the thrust coefficient derived from the vortex cylinder model in Eq. (16) can be applied to the momentum theory as a modification to account for this effect (Branlard and Gaunaa, 2015a). When the wake rotation effect is included in the momentum theory, the annulus axial induced velocity

as well as the tangential induced velocity from the momentum theory and the vortex cylinder model are identical. The radial induced velocity is available from the vortex cylinder model, but is not modelled in the momentum theory. However, for straight blades forming planar rotor, the radial induced velocity has no effect on the convergence calculation or the aerodynamic load calculation. This is because the radial velocity has no contribution to the projection of the velocity into the 2-D airfoil section for the straight, non-swept blades that are perpendicular to the rotor axis. These relationships can be written in the following

mathematical form, where the subscript of VC represents the vortex cylinder model and the subscript of MT represents the momentum theory, the superscript of pl represents the planar rotor.

$$u_{a,\mathrm{MT}} = u_{a,\mathrm{VC}}^{\mathrm{pl}} \tag{30a}$$

$$u_{t,\mathrm{MT}} = u_{t,\mathrm{VC}}^{\mathrm{pl}} \tag{30b}$$

$$u_{r,\mathrm{MT}} \equiv 0 \tag{30c}$$

### 3.5.2 Non-planar rotor

For a non-planar rotor, the inductions from the momentum theory with wake rotation effect included are equivalent to the inductions from the vortex cylinder model for the corresponding planar rotor and excluding the radial induced velocity. This means the momentum theory equivalently assumes the rotor is planar when calculating inductions. Then, the vortex cylinder model for the non-planar rotor is equivalent to the momentum theory with the following corrections: For the annulus axial

induced velocity, the correction is the difference of the results of the non-planar rotor and the corresponding planar rotor from the vortex cylinder model. The tangential induced velocity from both methods are the same, as described in Sect. 3.4.3. The radial induction is not available from the momentum theory, so the correction should be the complete radial induction of the non-planar rotor from the vortex cylinder model.





These relationships between the momentum theory and the vortex cylinder model for the non-planar rotor are summarized in the equations as follows, where the superscript of np represents the non-planar rotor.

$$u_{a,\mathrm{VC}}^{\mathrm{np}} = u_{a,\mathrm{MT}} + \Delta u_a \tag{31a}$$

$$u_{t,\mathrm{VC}}^{\mathrm{np}} = u_{t,\mathrm{MT}} \tag{31b}$$

$$u_{r,\mathrm{VC}}^{\mathrm{np}} = u_{r,\mathrm{MT}} + \Delta u_r \tag{31c}$$

where the corrections are:

$$\Delta u_a = u_{a,\mathrm{VC}}^{\mathrm{np}} - u_{a,\mathrm{VC}}^{\mathrm{pl}} \tag{32a}$$

$$\Delta u_r = u_{r,\mathrm{VC}}^{\mathrm{np}} \tag{32b}$$

## 4 Some important aspects in models using blade element theory

Some important aspects of the implementation of the low-fidelity models that use blade element theory and rely on the 2-D airfoil data are briefly discussed. They are important for the load calculation and to get good agreement with the higher-fidelity models.

### 4.1 Impact of unsteady airfoil aerodynamics on steady state

For the blade with out-of-plane shapes, it is necessary to include the unsteady airfoil aerodynamics model (usually referred to as the dynamic stall model), even for the steady-state simulation. Otherwise, the results of the tangential forces will have a visible error. The reason originates from the conclusions of unsteady 2-D aerodynamics: the correct circulatory lift can be obtained if the magnitude of the effective angle of attack is determined at the three-quarter-chord point, but the direction of it should be determined by the velocity at the quarter-chord point (Gaunaa, 2002; Bergami and Gaunaa, 2012). For the low-fidelity model using blade element theory, the angle of attack is usually determined at only one calculation point per section. For instance, for the BEM module in the HAWC2 code, it corresponds to the three-quarter-chord point. The direction of the circulatory lift is then equivalently transformed to perpendicular to the velocity at the quarter-chord point by including an additional torsion rate drag that is proportional to the circulatory lift. In addition, the non-circulatory part of the lift force should be correctly included. For the blade with out-of-plane shapes, even when the rotational speed is constant, the mid-chord point will experience a component of acceleration that is perpendicular to the chord due to the projection of centrifugal acceleration. In addition, the angular velocity vector will also have a projection in the 2-D section that will result in an effective pitching motion of the airfoil section. The contribution of both the mid-chord acceleration and the torsion rate should be included when calculating the aerodynamic force. For details, see the reports by Hansen et al. (2004) and Pirrung and Gaunaa (2018).





## 4.2 Curved blade length projection correction

For blades with in-plane or out-of-plane shapes, the curved blade length in an elementary annulus ($\mathrm{d}s$) is different from the change of the radius ($\mathrm{d}r$). Then, for the momentum analysis in a stream-tube, it is necessary to multiply the local thrust and torque coefficient with the term of $\frac{\mathrm{d}s}{\mathrm{d}r}$ to account for this difference (Madsen et al., 2020a). For the Kutta-Joukowski analysis

of the non-planar rotor in Eq. (5), the term of $\frac{\mathrm{d}s}{\mathrm{d}r}$ is already included in the equation of force. So, it is not necessary to include this term again in the thrust coefficient in Eq. (22) during the convergence calculation using the vortex cylinder model.

## 5 Blade element theory with vortex cylinder model

The BEM method is the blade element theory coupled with the momentum theory. Similarly, the vortex cylinder model should be coupled with the blade element theory for the aerodynamic load calculation of a rotor with finite number of blades. The

link between the blade element theory and the vortex cylinder method is the relationship of the blade bound circulation and the trailed vorticity strength of the vortex cylinder. The trailed vorticity strength is calculated from the total bound circulation of the two neighbouring sections using Eq. (24). The total bound circulation is calculated from the blade bound circulation and assuming all blades have the same bound circulation strength.

$$\Gamma_i = N_B \Gamma_i^B \tag{33}$$

The blade bound circulation is calculated from the circulatory part of the lift coefficient $C_L^C$. For quasi-steady simulations, the circular part of the lift coefficient can be replaced by the quasi-steady lift coefficient $C_L^{QS}$.

$$\Gamma^B = \frac{1}{2} V_{rel} c C_L^C \tag{34}$$

In this section, the coupling of the vortex cylinder model and the blade element theory will be firstly described in Sect. 5.1. The application of the tip-loss factor for the non-planar rotor is important, and is described in Sect. 5.2. Finally, the implemen-

tation of the proposed vortex cylinder model is summarized in Sect. 5.3.

## 5.1 Vortex cylinder model as a correction to BEM or as the full model

According to the relationship between the momentum theory and the vortex cylinder model described in Sect. 3.5, there are two possible methods of using the vortex cylinder model when coupling with the blade element theory. Firstly, the vortex cylinder model can be used as a modification to the existing BEM model, which is named and labelled as BEM-VC model. Otherwise,

the vortex cylinder model can be considered as the complete replacement of the BEM method. With the blade element theory coupled with the vortex cylinder model, we have the blade element vortex cylinder (BEVC) model, which does not include any momentum theory results. The reader may argue that the BEVC model should completely replace the BEM model for the prediction of the aerodynamic loads in an existing aeroelastic code. As a first step, the authors recommend the use of the vortex cylinder model as a correction to the BEM method (BEM-VC). This is because the framework of the BEM method with many

sub-models and corrections has been implemented in the aeroelastic codes and has been intensively tested.





## 5.2 Tip-loss correction

Prandtl's tip-loss factor is commonly applied to the BEM method to account for the difference between a finite number of blades and the assumption of infinite number of blades in the momentum theory (Glauert, 1935; Sørensen, 2015). Similarly, the vortex cylinder model also assumes infinite number of blades. Also, considering the relationship between the momentum
theory and the vortex cylinder model discussed in the previous sections, a tip-loss correction should be applied to the vortex cylinder model. The tip-loss factor $F$ presented by Glauert (1935) was implemented in the BEM module in the HAWC2 code (Madsen et al., 2020a):

$$F = \frac{2}{\pi} \cos^{-1} \left( \exp \left( -\frac{N_B}{2} \frac{R_{\text{tot}} - r}{r \sin \varphi} \right) \right) \tag{35}$$

where $\varphi$ is the inflow angle.

The tip-loss correction is applied by scaling the thrust coefficient with the inverse of the tip-loss factor when calculating the blade axial induction.

$$a_B^{\text{pl}} = f_{a-C_T}(C_{T,\text{eff}}/F) \tag{36}$$

where the subscript $B$ indicates the induction at the blade.

### 5.2.1 Tip-loss for non-planar rotor

Care should be taken when applying the tip-loss correction to the non-planar rotor. The first aspect is the angle to use when calculating the tip-loss factor in Eq. (35). The inflow angle in the rotor coordinate system, which is the flow angle seen by the rotor plane, is usually used (Madsen et al., 2020a). Another possible choice is the flow angle in the sectional coordinate system, which is the flow angle seen by the 2-D airfoil section. For planar rotors, it is not necessary to distinguish between them because they are identical. Since the tip-loss factor is developed for planar rotors, it is not possible to analytically show which
one is better than the other when applied to non-planar rotors. In a preliminary numerical investigation that is not reported in the present work, it was discovered that results when using the sectional flow angle to calculate the tip-loss factor are in slightly better agreement with the higher-fidelity models compared to using the inflow angle. As a result, the sectional flow angle is recommended when calculating the tip-loss factor, and is used in the BEM-VC method when calculating the results in Sect. 7.

The second aspect is that the tip-loss factor is only to model the amplified axial induction at the blade compared to the
annulus-averaged axial induction, it should not directly change the trailed (tangential or longitudinal) vorticity strength of the vortex cylinders. Recall the similarity of the vortex cylinder model for the non-planar rotor and the corresponding planar rotor with the same circulation distribution described in Sect. 3.4. The correct implementation of the tip-loss correction in the vortex cylinder model for the non-planar rotor could be considered as a two-step approach.

In the first step, the axial induction factor at the blade of the corresponding planar rotor with the tip-loss correction is
calculated using Eq. (36). The second step is to calculate the difference between the annulus axial induction of the non-planar rotor and the planar rotor using the vortex cylinder model. From the effective thrust coefficient, the annulus-averaged axial





induction factor of the planar rotor is calculated.

$$a_\infty^{\text{pl}} = f_{a-C_T}(C_{T,\text{eff}}) \tag{37}$$

where the subscript $\infty$ represent infinite number of blades or annulus-averaged value.

The tangential vorticity of the vortex cylinder is calculated from the annulus axial induction of the planar rotor $a_\infty^{\text{pl}}$ using
Eq. (20) and is duplicated here with the update notation.

$$\gamma_{t,i}^{\text{np}} = \gamma_{t,i}^{\text{pl}} = 2U_0(a_{\infty,i}^{\text{pl}} - a_{\infty,i-1}^{\text{pl}}) \tag{38}$$

Then, the annulus axial induction factor of the non-planar rotor $a_\infty^{\text{np}}$ can be calculated using Eq. (9). Finally, the axial induction at the blade section $i$ of the non-planar rotor is then equal to the sum of the blade axial induction of the planar rotor and the difference of the annulus axial induction of the non-planar rotor and the planar rotor.

$$a_B^{\text{np}} = a_B^{\text{pl}} + a_\infty^{\text{np}} - a_\infty^{\text{pl}} \tag{39}$$

The tip-loss factor is only applied to the axial induction but not applied to the tangential or radial induction, which is following the application of the tip-loss correction in the BEM module in the HAWC2 code.

### 5.2.2  Erroneous implementation

If the model is used without clearly distinguishing between the axial induction on the blade and the annulus-averaged axial
induction, the resulting system closure could be wrong. If using the blade axial induction factor $a_B^{\text{pl}}$ instead of the annulus axial induction factor $a_\infty^{\text{pl}}$ to calculate tangential vorticity in Eq. (38), the tangential vorticity will be directly scaled by the tip-loss factor, which is unphysical. Then, the annulus axial induction and the radial induction calculated using Eqs. (9) and (10) will be directly scaled by the tip-loss factor due to the wrong tangential vorticity.

For the planar rotor with straight blades, the calculated aerodynamic loads on the blade using the erroneous method will
still be correct. The tangential vorticity from the erroneous method is wrong and the radial induction calculated using Eq. (10) will then be wrong. However, the blade axial induced velocity and the tangential induced velocity are correctly calculated. In addition, the radial induction has no contribution to the aerodynamic loads because it has no contribution to the flow seen by the 2-D section when the blade is straight and the rotor is planar, as described in Sect. 3.5.1.

### 5.2.3  Other implementation of tip-loss correction

The tip-loss correction used in the present work is scaling the thrust coefficient when calculating the blade axial induction and is actually only applied to the planar part of the axial induction. There are other definitions of the tip-loss factor, such as the ratio of the blade axial induction and the annulus-averaged axial induction (Branlard and Gaunaa, 2014). Then, it is possible to directly utilize the tip-loss factor as:

$$F = \frac{a_\infty^{\text{np}}}{a_B^{\text{np}}} \tag{40}$$





However, consider that the tip-loss factor is originally developed for planar rotors. As in Prandtl's simple model of system of material sheets, the flow will go around the vortex disc edges. Also for the modern definition of the tip-loss factors (Branlard and Gaunaa, 2014), planar rotor disc is always assumed. Then, the method of directly applying Prandtl's tip-loss factor in Eq. (35) to the non-planar axial induction as in Eq. (40) is then without a clear physical background and thus not recommended.

## 5.3   Algorithm of proposed vortex cylinder models

As has been described previously in this section, the proposed vortex cylinder model can be used in two ways: either as a correction to the BEM method (BEM-VC) or is solely used and coupled with the blade element theory (BEVC). Details of the implementation of both methods have been described previously in this work, and are summarized in Algorithm 1.





---

**Algorithm 1** Overview of the proposed BEM-VC / BEVC method

---

**for** each time step **do**

    **for** each blade section **do**

        Calculate the angle of attack $\alpha$ at the three-quarter-chord point using:

            section velocity, deflection, pitch angle, torsion rate, twist, induced wind and free wind

        Compute the quasi-steady lift coefficient $C_L^{QS}$ and drag coefficient $C_D^{QS}$ using the airfoil look-up table

        **BEVC model:**

            Calculate the bound circulation strength $\Gamma^B$, Eq. (34)

            Calculate the Kutta-Joukowski thrust coefficient $C_{T,\text{KJ}}$, Eq. (22)

        **BEM-VC model:**

            Compute the curved blade length projection correction $\frac{\mathrm{d}s}{\mathrm{d}r}$ (Madsen et al., 2020a, Eq. (32))

            Calculate the thrust coefficient $C_{T,\text{MT}}$ and torque coefficient $C_Q$ of each stream tube by projecting the lift and possibly the drag

        with respect to the rotor plane, see Madsen et al. (2020a). The $\frac{\mathrm{d}s}{\mathrm{d}r}$ term should be included

        **Comment:** $C_{T,\text{MT}} = C_{T,\text{KJ}}$ when drag is excluded in the momentum balancing

    **end for**

    **for** each blade section, from most outboard towards inboard **do**

        **Optional:** Compute thrust coefficient due to wake rotation, Eq. (16)

        Compute the effective thrust coefficient of the annulus $C_{T,\text{eff}}$, Eq. (14)

        Divide $C_{T,\text{eff}}$ by the tip-loss factor $F$ in Eq. (35) to get the scaled thrust coefficient: $C_{T,\text{eff}}/F$

        **BEVC model:**

            Compute the blade axial induction factor of the corresponding planar rotor $a_B^{\text{pl}}$ using the scaled thrust coefficient $C_{T,\text{eff}}/F$, Eq. (21)

        **BEM-VC model:**

            Compute the blade axial induction factor $a_{\text{BEM}}$ as in BEM method, using the scaled thrust coefficient $C_{T,\text{eff}}/F$, Eq. (21)

        **Comment**: $a_{\text{BEM}} = a_B^{\text{pl}}$ when drag is excluded in the momentum balancing

        Compute the annulus induction factor of the corresponding planar rotor $a_\infty^{\text{pl}}$ using the thrust coefficient $C_{T,\text{eff}}$, Eq. (21)

        Compute the strength of the tangential vorticity $\gamma_t$ using $a_\infty^{\text{pl}}$, Eq. (38)

        Compute the annulus axial induction of the non-planar rotor $a_\infty^{\text{np}}$, based on Eq. (9)

        Compute the blade axial induction of the non-planar rotor $a_B^{\text{np}}$, Eq. (39)

        Compute the tangential induced velocity $u_t$, Eq. (29)

        Compute the radial induced velocity $u_r$, Eq. (10)

    **end for**

    **for** each blade section **do**

        Compute aerodynamic forces including dynamic stall and Theodorsen effects (Hansen et al., 2004; Pirrung and Gaunaa, 2018)

    **end for**

**end for**

---





# 6 The models for comparison

The higher-fidelity models for the comparison are the Navier-Stokes solver EllipSys3D (Michelsen, 1992, 1994; Sørensen, 1995) and the lifting-line module in the aerodynamic solver MIRAS (Ramos-García et al., 2016), both developed at the Technical University of Denmark (DTU). The lower-fidelity aerodynamic models used for comparison are the BEM method, the
BEM method with radial induction correction by Madsen et al. (2020a) (BEM-$u_r$) and the proposed BEM-VC method. Details of these model setups are given in this section.

## 6.1 Navier-Stokes solver

The pressure-based incompressible three-dimensional solver EllipSys3D was used to solve the Reynolds-Averaged Navier-Stokes equations, using a finite volume discretization. An inlet/outlet strategy was followed for the boundary conditions of
the outer limit of the CFD domain. The flow was assumed to be fully turbulent, and the k-$\omega$ SST model (Menter, 1994) was employed. These higher-fidelity simulations are labelled in the present work as CFD.

Several rotor-resolved meshes were built. They were generated in two consecutive steps, that were fully scripted in order to ensure a similar resulting grid quality. First, a structured mesh of the blade surface was generated with the openly available Parametric Geometry Library (PGL) tool (Zahle, 2019). A total of 128 cells were used in the spanwise direction, and the
chordwise direction was discretized with 256 cells. Secondly, the surface mesh was radially extruded with the hyperbolic mesh generator Hypgrid (Sørensen, 1998) to create a volume grid. A total of 256 cells were used in this process, and the resulting outer domain was located at approximately 11 rotor diameters. A boundary layer clustering was taken into account, with an imposed first cell height of $1 \times 10^{-6}$ m, in order to target $y^+$ values lower than unity. The resulting volume meshes accounted for a total of 14.2 million cells. The grid topology is illustrated in Fig. 6, through the particular case of the baseline straight
blade.

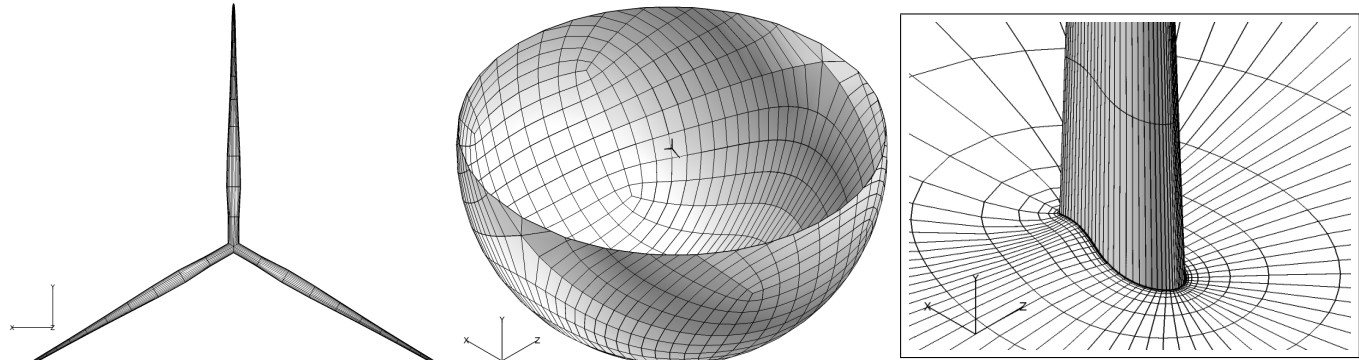

**Figure 6.** Visualization of straight blade CFD mesh. Left: blade surface mesh (for clarity, only 1 out of 16 grid lines shown). Middle: lower half of outer domain surface mesh (1 out of 16 grid lines). Right: detail of volume mesh, cut at mid-span (1 out of 8 grid lines).



While a steady solver was used, unsteady separation is expected near the root of the wind turbine blade in operation. To mitigate the effects that this can have on the conclusions of the present work, all the CFD quantities were averaged for the last 350 iterations.

## 6.2 Lifting-line solver

The lifting-line module in the aerodynamic solver MIRAS (Ramos-García et al., 2016) is implemented as a time-marching approach and uses the 2-D airfoil data. This work uses a modified version of the lifting-line module that includes the influence of the curved bound vortex on the induced velocity as described by Li et al. (2020) which is labelled as LL-mod in that work. The bound vorticity is located at the quarter-chord line and the calculation points are placed on the three-quarter-chord line. The influence of the curved bound vortex is modelled by adding the difference of the induced velocity due to the 3-D bound vorticity and an imaginary 2-D bound vorticity (infinitely long line vortex) evaluated at the three-quarter-chord point to the induced velocity of the blade section (Li et al., 2020). The curved bound vortex influence is assumed to be constant along the chord. The angle of attack to determine the circulatory lift and drag coefficient is the angle of attack at the three-quarter-chord point. The flow environment to determine lift and drag, especially the direction of the lift and drag force, is at the quarter-chord point. In addition, as the discussion in Sect. 4.1, the non-circulatory lift is included in the lifting-line model when calculating the aerodynamic forces. For the setup in this study, each blade is discretized into 50 sections with cosine spacing. Each simulation is calculated for 20 thousand time steps and each step correspond to $1.5°$ of azimuthal angle, resulting in a total of 83.3 revolutions. The airfoil data is from 2-D fully turbulent CFD results (Bortolotti et al., 2019). The vortex core size in the calculations is 0.1% of the local chord length. The first row of trailed vorticities begins from the trailing edge of the blade.

## 6.3 Low-fidelity models

Three low-fidelity aerodynamic models are used for the comparison. The first one is the BEM method implemented in the HAWC2 code version 12.8 (Larsen and Hansen, 2007). The second one is the BEM method with radial induction correction in Eq. (6) (BEM-$u_r$) in the same version of HAWC2 code, and is described by Madsen et al. (2020a). The third one is the BEM-VC method proposed in this work and is implemented in a test version of the HAWC2 code based on version 12.8. As has been discussed in Sect. 5.1, the proposed BEM-VC method utilizes the vortex cylinder model as a correction to the existing BEM method. The results should be identical to the BEVC model when the drag is excluded in the momentum balancing. So, the results from the BEVC model, which solely uses the vortex cylinder model and not directly uses the momentum theory, are not shown. For the low-fidelity models in the HAWC2 code, each time step corresponds to 0.01 s and each simulation is calculated for 700 s to get the steady-state value. Each blade is discretized radially into 80 sections. The airfoil data is also from 2-D fully turbulent CFD results and is identical to those used in the lifting-line method. As has been described in Sect. 5.2, the flow angle seen by the airfoil section is used to calculate the tip-loss factor for the BEM-VC method. For the BEM method and BEM-$u_r$ method, the original implementation of the tip-loss factor in the HAWC2 code using the inflow angle is





applied. Since all three low-fidelity models are implemented in the HAWC2 code, it is then guaranteed the transformation of the velocity and force between different coordinate systems during the computation is consistent.

## 7 Results

In this section, the distributed aerodynamic load in the out-of-plane and in-plane direction, as well as the integrated loads
of aerodynamic thrust and power from different low-fidelity models, are compared with results from higher-fidelity models. The higher-fidelity models are the Navier-Stokes solver (CFD) and the lifting-line method (LL) as described in Sect. 6. By comparing the results from the three low-fidelity models with higher-fidelity models, it will be highlighted to which extent the influence of the non-planar rotor geometry can be correctly modelled by each of the lower-fidelity models.

### 7.1 Test cases

There are five different wind turbine blades used for the comparison, all of them are based on the IEA-10.0-198 10 MW reference wind turbine (RWT) (Bortolotti et al., 2019). The baseline straight blade is modified by aligning the half-chord line to a straight main-axis. For the upwind dihedral blades, the main-axes that determine the planforms are obtained from modified Bézier curves which are parameterized with: dihedral ratio $\bar{r}_s$, dihedral magnitude $\Delta d$ and tip dihedral angle $\Lambda_{tip}$. In addition, some cases with large cone angles are applied to these dihedral blades to exploit the range of capability of the models. The
radius of the unconed rotor is 99 m, of which the hub radius is 2.8 m. The parameterization of the dihedral blades is very similar to that for the previous study of swept blades (Li et al., 2018). The dihedral blades W-1 to W-4 are corresponding to Blade-1 to Blade-4 in the previous study, but with out-of-plane shapes (dihedral) instead of in-plane shapes (sweep). The blades are assumed to be stiff, which means the effect of elastic deformation is not included. In addition, the pitch angle is zero for all test cases.

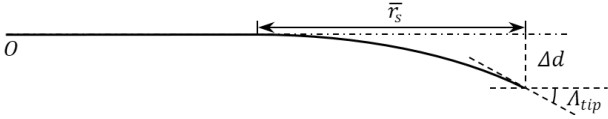

**Figure 7.** The parameterization of the dihedral blade with dihedral ratio $\bar{r}_s$, dihedral magnitude $\Delta d$ and tip dihedral angle $\Lambda_{tip}$. The figure is from (Li et al., 2018) but the definition of the parameters are modified.

The parameters of these upwind dihedral blades are summarized in Table 1. The main-axes of these dihedral blades are illustrated in Fig. 7. The purpose of having dihedral blades with different dihedral magnitude and different tip dihedral angle is to represent different possible shapes of the dihedral blades.

The airfoils are aligned perpendicular to the curved main-axis, which is the half-chord line. The chord and twist distribution of the dihedral blades remain unchanged compared to the baseline straight blade. The radius of the dihedral blades are identical
to that of the baseline straight blade, but the curved blade length is increased due to the dihedral. For the simulations in this





section, the uniform inflow of $8\,\mathrm{m\,s^{-1}}$ with no yaw error is applied to the rotor with a constant rotational speed of $0.855\,\mathrm{rad\,s^{-1}}$. For the unconed rotors, the tip-speed ratio is 10.58.

**Table 1.** The parameters of the planforms of the four upwind dihedral blades used for the comparison.

| Name | Dihedral ratio $\bar{r}_s$ | Dihedral magnitude $\Delta d$ | Tip dihedral angle $\Lambda_{tip}$ |
|------|------|------|------|
| W-1 | 50% | 10% | 20° |
| W-2 | 50% | 10% | 40° |
| W-3 | 25% | 5% | 20° |
| W-4 | 25% | 5% | 40° |

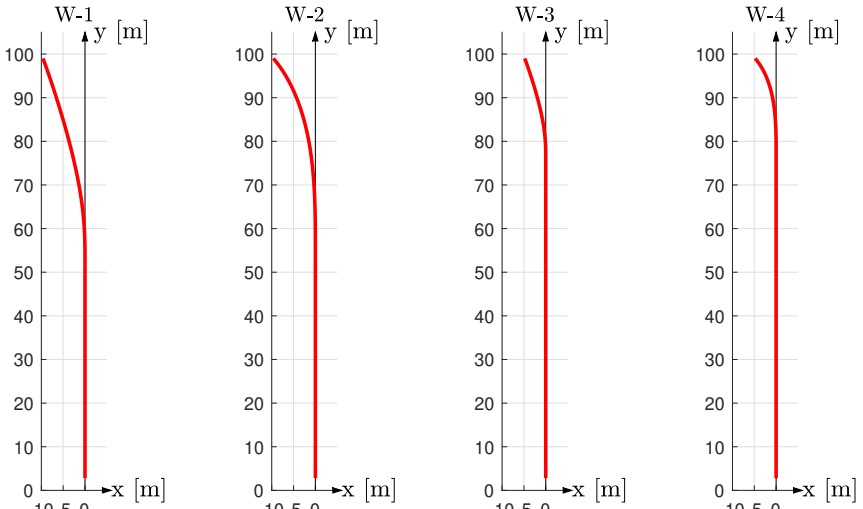

**Figure 8.** Side-view of the main-axes of the four different upwind dihedral blades used for the comparison. The dihedral blades from left to right are W-1 to W-4.

## 7.2 The distributed load

For the test cases described in Sect. 7.1, the distributed aerodynamic loads calculated from different aerodynamic models are summarized and are compared in this section. For the calculation of the aerodynamic loads, both lift and drag force are included. In this study, the focus is on the influence of blade dihedral on the loads. The near-root region (i.e. up to an approximate radius of 20 m), experienced flow separation in the CFD solution, and it is not the focus of this study.





### 7.2.1 Baseline blade with zero cone

Firstly, the steady-state results of the baseline straight blade without cone calculated from different models are compared and plotted in Fig. 9. Please note that the distributed loads plotted from all models are corresponding to aerodynamic force per unit radius. The three BEM methods give identical results as expected. The results from the higher-fidelity models are similar to the results from the BEM models.

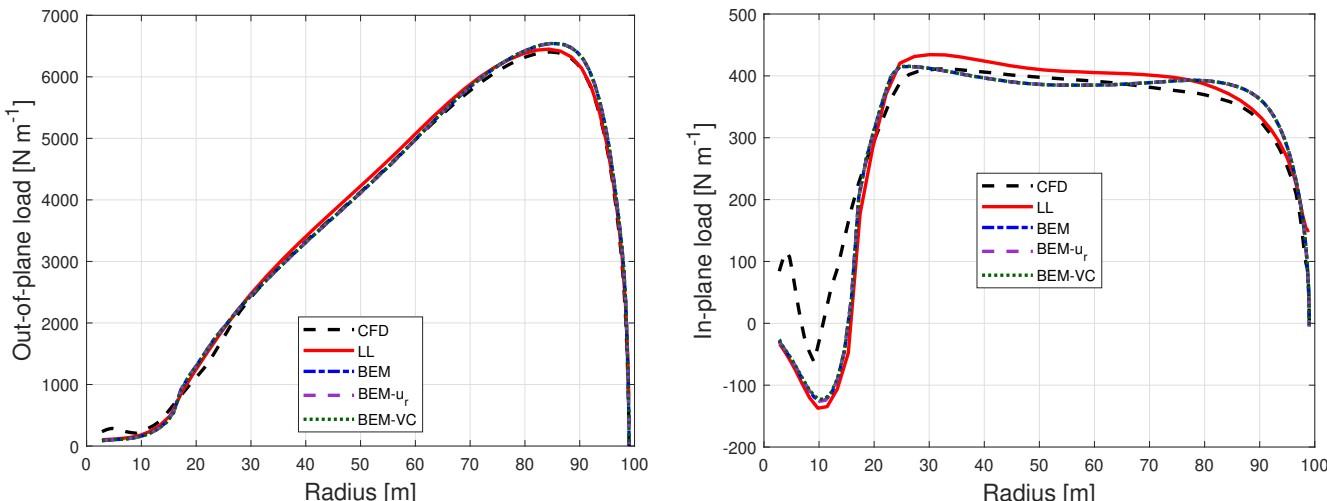

**Figure 9.** Comparison of out-of-plane load (left) and in-plane load (right) of the baseline straight blade calculated from CFD, lifting-line method (LL), BEM method, BEM with radial induction (BEM-$u_r$) and the proposed BEM-VC method. The results from three BEM variants coincide with each other as expected.

### 7.2.2 Dihedral blade with zero cone

The steady-state results of the different upwind dihedral blades with zero cone angle are calculated with different aerodynamic models. The out-of-plane load and in-plane load of the dihedral blade W-1 is shown in Fig. 10. The distributed loads correspond to aerodynamic force per unit radius, the curved blade length projection correction described in Sect. 4.2 is applied.

Comparing the distributed load of the baseline straight blade and the dihedral blade W-1, it is difficult to draw conclusions for the out-of-plane load or the in-plane load, because no clear trends can be seen. In order to clearly show the influence of the dihedral on the loads predicted by different aerodynamic models, the difference of the loads of the dihedral blade W-1 with respect to the baseline straight blade are shown in Fig. 11. This is done by directly subtracting the load of the baseline straight blade from the load of the dihedral blade for the same radius. Consistently to this remark, the following comparisons

throughout this work only include the difference of the distributed loads. The absolute loads are not shown since it is more difficult to draw conclusions from them.

    It can be seen that for the difference of both the out-of-plane and in-plane load, both higher-fidelity models (CFD and LL) predict a fairly similar pattern of spanwise load redistribution. For the spanwise location that is further inboard compared to



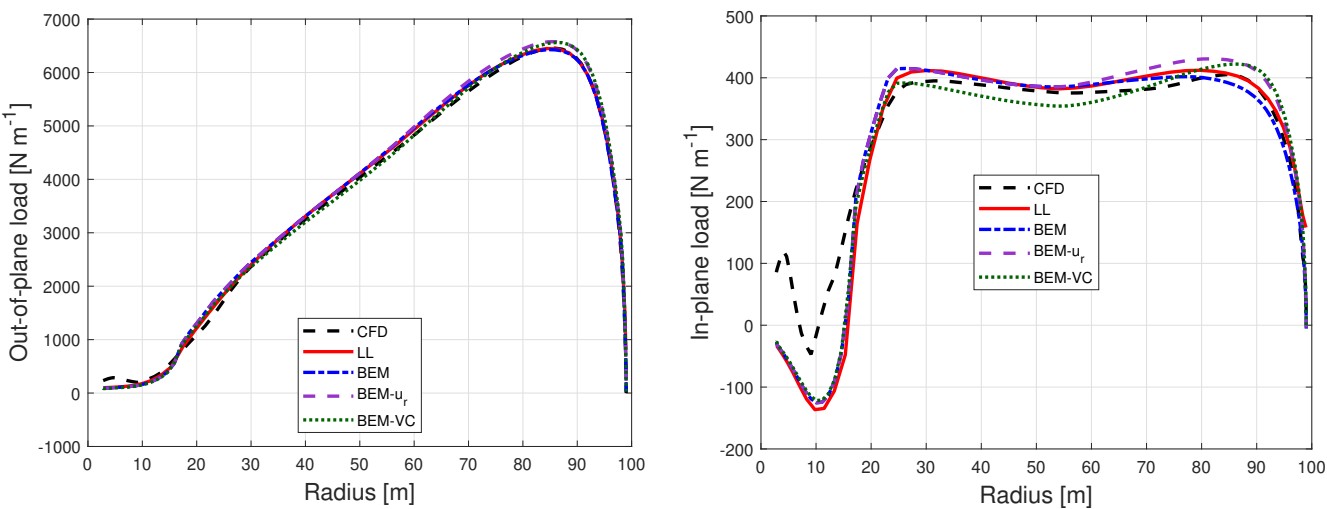

**Figure 10.** Comparison of out-of-plane load (left) and in-plane load (right) of the dihedral blade W-1 calculated from CFD, lifting-line method (LL), BEM method, BEM with radial induction (BEM-$u_r$) and the proposed BEM-VC method.

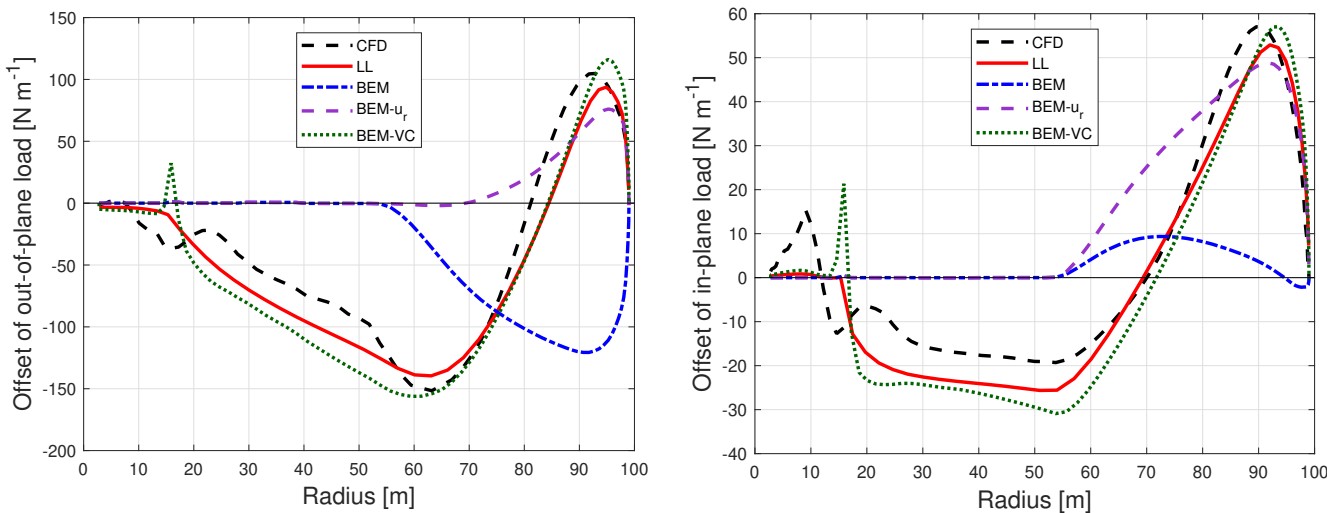

**Figure 11.** Comparison of the difference of the out-of-plane load (left) and in-plane load (right) of the dihedral blade W-1 compared to the baseline blade calculated from CFD, lifting-line method (LL), BEM method, BEM with radial induction (BEM-$u_r$) and the proposed BEM-VC method.





where the blade starts to dihedral, the out-of-plane and in-plane loads of the upwind dihedral blade are lower compared to the baseline straight blade. When moving from the spanwise location where the blade starts to dihedral towards halfway until the blade tip, both out-of-plane and in-plane loads are also lower compared to the baseline. When moving further towards the tip, both out-of-plane and in-plane loads are then increased compared to the baseline until the blade tip.

A similar pattern of spanwise load redistribution was also observed for swept blades in the previous works (Li et al., 2018, 2020, 2021). However, the redistribution of the loads only takes place where the blade is swept. The loads of the swept blade and the straight blade are almost identical for the inboard part of the blade, where the blade is still straight. Instead, for the dihedral blade, the influence of the blade dihedral has a pronounced influence on the inboard part of the blade that is still straight. This means the blade dihedral at the outboard part of the blade has an influence throughout the spanwise locations of 

the blade instead of only on the part of the blade that has dihedral.

For both the ordinary BEM method and the BEM method with radial induction correction (BEM-$u_r$), the influence of blade dihedral is not correctly predicted. For the inboard part of the blade that has no dihedral, both methods predict zero offset of loads. The BEM method predicts lowered out-of-plane load for the entire portion of the blade that has dihedral. For the in-plane load, the BEM method predicts neglectable differences compared to the baseline straight blade. The performance of

the BEM method is as expected because of the assumption of radially independence in the stream tube theory and the changed streamwise starting position of the trailed vorticity due to blade dihedral is not modelled. The BEM-$u_r$ method is able to predict the increase of the load near the tip, for both out-of-plane and in-plane loads. However, the decrease of the loads near where the blade starts to dihedral and also further inboard is not predicted. Because for the BEM-$u_r$ method, the radial dependency is modelled to a limited extent and is only on the radial induction but not on the axial induction (Madsen et al., 2020a). In

addition, the axial and radial induction from the BEM-$u_r$ method corresponds to a planar rotor.

In comparison, the proposed BEM-VC method correctly predicts the pattern and the magnitude of the load redistribution for both out-of-plane and in-plane loads. The decrease of the loads further inboard compared to where the blade starts to dihedral is also well predicted. It should be highlighted that the spanwise location of the crossing of the zero load difference is also in good agreement with the results from higher-fidelity models (CFD and LL). The largest error with the BEM-VC method is

mainly for the tip-most part: the increase of the load is over-predicted for both out-of-plane and in-plane loads. This could be due to the use of Prandtl's tip-loss correction in the model. The authors believe that a more advanced aerodynamic model that can replace the current tip-loss correction, if coupled with the proposed vortex cylinder model, could have better agreement with higher-fidelity models. An example is the near-wake model (Madsen and Rasmussen, 2004; Pirrung et al., 2017) which approximately models the near-wake as the helical trailed vorticity and is currently coupled with a far-wake model that is based

on the momentum theory.

The results of the other three upwind dihedral blades are shown in Appendix B1. The same conclusions as that for the blade W-1 also hold for these results.



### 7.2.3  Upwind cone

To exploit the range of validity of the proposed method, a large upwind cone of $15°$ is applied to the baseline straight blade as well as the blades with upwind dihedral (W-1 to W-4). For the coned cases, the radius of the rotor will decrease compared to the radius of the rotor having the same blades but with zero cone. For better comparison, the abscissa in the figures correspond to the radius of the blade without cone, and the loads are defined as force per unit radius. The factor of $\frac{ds}{dr}$, which now equals to the secant of the sum of the cone angle ($\theta_c$, positive when cone up-wind) and the dihedral angle $\kappa$, is multiplied to the loads of the coned blades.

$$\frac{ds}{dr}(r_i) = \frac{1}{\cos(\kappa_i + \theta_c)} \tag{41}$$

For the baseline straight blade with $15°$ of cone upwind, the difference of the loads compared to the straight blade without cone is plotted in Fig. 12. For the upwind dihedral blades with $15°$ of cone further upwind, the difference of the loads compared to the baseline blade with the same upwind cone angle are calculated. The results of the upwind coned dihedral blade W-1 is shown in Fig. 13 in this section. The results of the other dihedral blades with further upwind cone are summarized in Appendix B2.

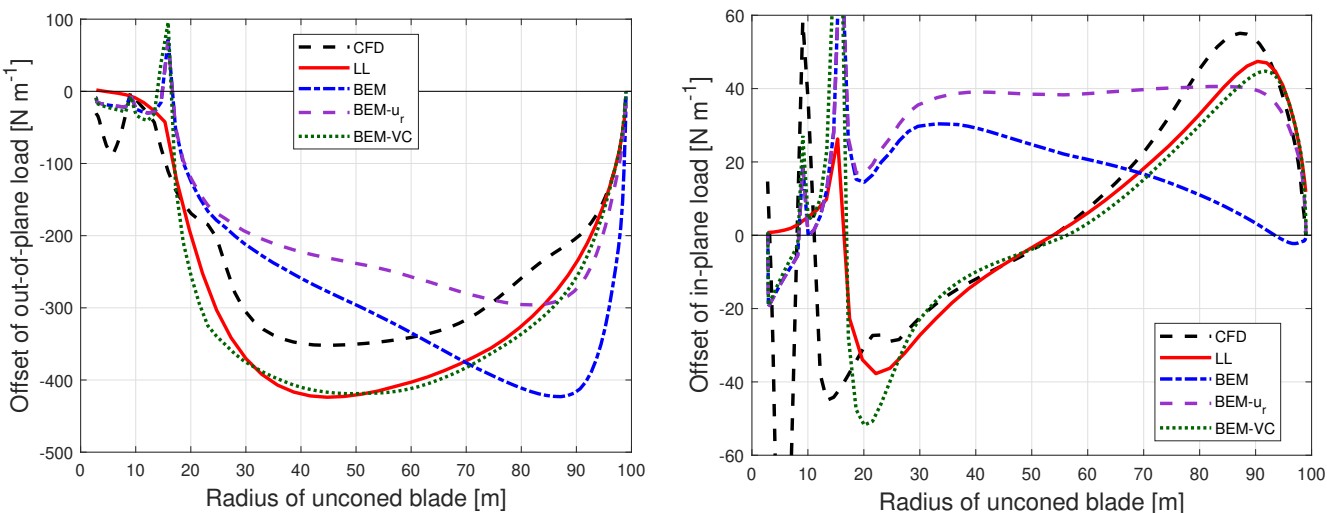

**Figure 12.** Comparison of the difference of the out-of-plane load (left) and in-plane load (right) of the baseline blade with $15°$ of cone upwind compared to the baseline blade without cone calculated from CFD, lifting-line method (LL), BEM method, BEM with radial induction (BEM-$u_r$) and the proposed BEM-VC method.

For the straight blade with large cone in Fig. 12, the results from the proposed method are in good agreement with the results from higher-fidelity models (CFD and LL). The proposed method predicts the same trends as the higher-fidelity models while the BEM method and BEM-$u_r$ method predict different trends. For the upwind dihedral blades with large cone angle in Fig. 13 and in Appendix B2, the results from the proposed method are also in better agreement with the higher-fidelity model compared to the BEM method and BEM-$u_r$ method. However, the results from the proposed method are having some





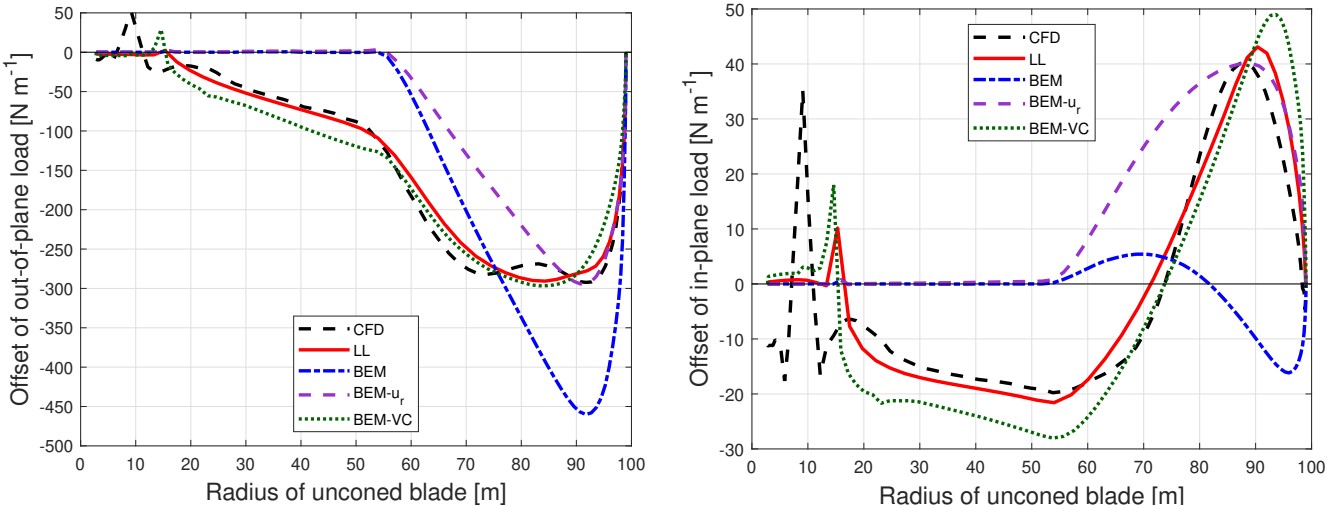

**Figure 13.** Comparison of the difference of the out-of-plane load (left) and in-plane load (right) of the dihedral blade W-1 with $15°$ of cone upwind compared to the baseline blade with the same cone calculated from CFD, lifting-line method (LL), BEM method, BEM with radial induction (BEM-$u_r$) and the proposed BEM-VC method.

differences compared to the higher-fidelity model probably due to the limitation of the current tip-loss correction, especially near the blade tip that has large blade dihedral.

### 7.2.4 Downwind cone

To further exploit the range of validity of the proposed method, a large downwind cone angle of $15°$ is applied to the baseline
straight blade as well as the blades with upwind dihedral (W-1 to W-4). As has been discussed, the radius of the coned rotor will change compared to the radius of the rotor with the same blades but without cone. For better comparison, the abscissa of the figures correspond to the radius of the blade without cone, and the loads are again defined as force per unit radius. The term of $\frac{\mathrm{d}s}{\mathrm{d}r}$ calculated from Eq. (41) should be multiplied to the loads of the coned blades. For the downwind-coned straight blade, the difference of the loads compared to the baseline straight blade without cone is plotted in Fig. 14.
For the upwind dihedral blades with $15°$ of cone downwind on top of it, the difference of the loads compared to the baseline blade with the same cone angle downwind are calculated. The results of the coned dihedral blade W-1 is shown in Fig. 15 in this section. The results of the other upwind dihedral blades with downwind cone are summarized in Appendix B3.

It can be seen that for both the straight blade and the upwind dihedral blade with large downwind cone, the results from the proposed method (BEM-VC) are in good agreement with the results from the higher-fidelity models (CFD and LL). On the
other hand, the BEM method and the BEM method with radial induction correction (BEM-$u_r$) are not able to correctly predict the difference of the loads.



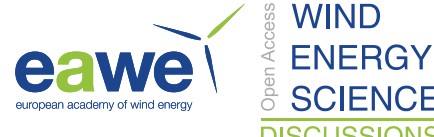

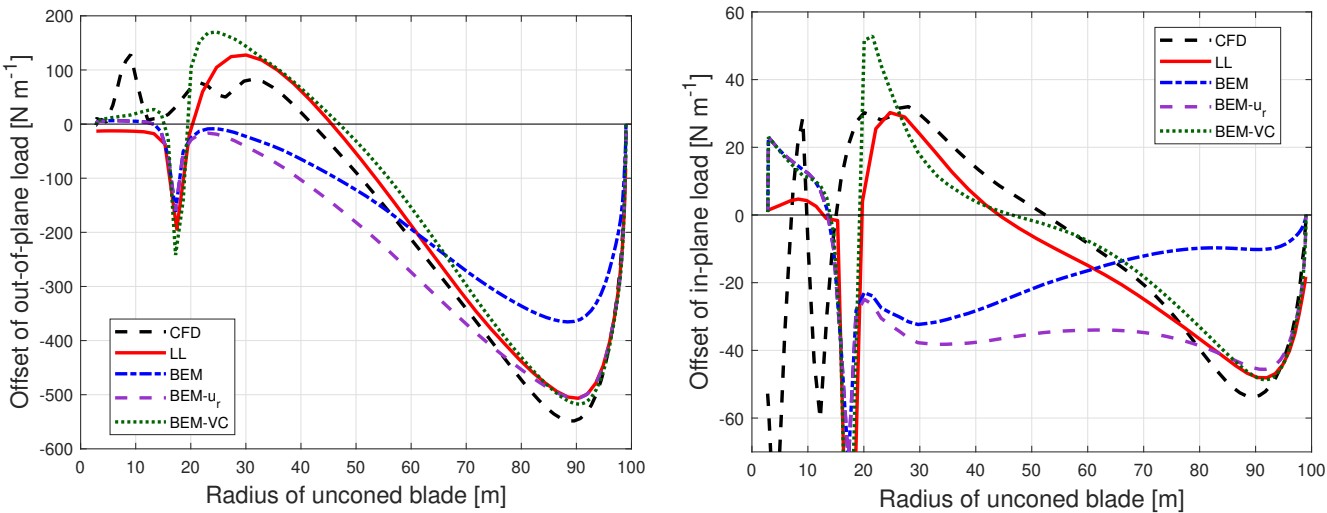

**Figure 14.** Comparison of the difference of the out-of-plane load (left) and in-plane load (right) of the baseline blade with 15° of cone downwind compared to the baseline blade without cone calculated from CFD, lifting-line method (LL), BEM method, BEM with radial induction (BEM-$u_r$) and the proposed BEM-VC method.

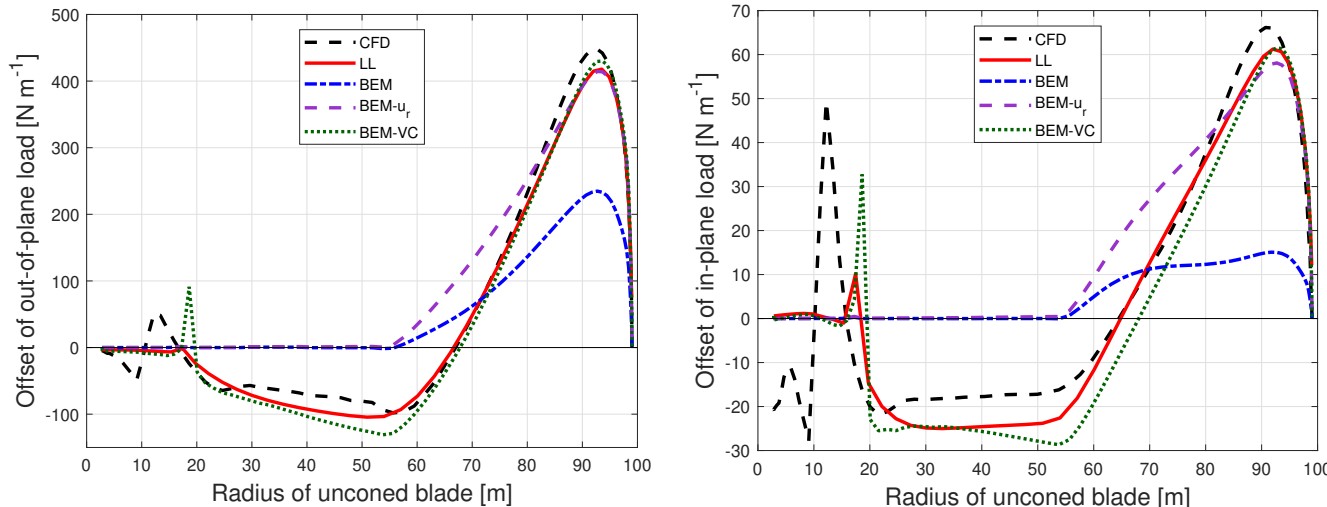

**Figure 15.** Comparison of the difference of the out-of-plane load (left) and in-plane load (right) of the upwind dihedral blade W-1 with 15° of cone downwind compared to the baseline blade with the same cone calculated from CFD, lifting-line method (LL), BEM method, BEM with radial induction (BEM-$u_r$) and the proposed BEM-VC method.



### 7.3 The integrated aerodynamic loads

The integrated aerodynamic loads, which are the aerodynamic power and thrust from different models are compared in this section. Please note that when comparing the integrated aerodynamic loads, errors in the distributed loads may cancel out. So, it is important to bear in mind that the performance of the different aerodynamic models is not fully represented by their

abilities to predict the total aerodynamic power or thrust of the rotor. The aerodynamic force (per unit length of radius) $\boldsymbol{F^*}$ on each blade section is consisted of the out-of-plane force $F_{oop}^*$, the in-plane force $F_{ip}^*$ and the radial force $F_{rad}^*$. They are defined to be positive when aligned with $x-$, $z-$ and $y-$coordinate respectively. For the calculation of the aerodynamic load, both lift and drag force are included. The position of applying the force on the blade section is $\boldsymbol{p}$. For simplicity, we use the half-chord point coordinate as $\boldsymbol{p}$. This means we neglect the distance between the half-chord point and the quarter-chord point

and also the contribution of the twist to the vector $\boldsymbol{p}$. In addition, the contribution of the sectional airfoil aerodynamic moment (calculated from $C_m$) to the aerodynamic momentum of the rotor is also neglected. Then, the distributed aerodynamic moment from each blade section is:

$$\boldsymbol{M} = \boldsymbol{p} \times \boldsymbol{F^*} = \begin{pmatrix} x \\ y \\ 0 \end{pmatrix} \times \begin{pmatrix} F_{oop}^* \\ F_{rad}^* \\ F_{ip}^* \end{pmatrix} = \begin{pmatrix} yF_{ip}^* \\ -xF_{ip}^* \\ xF_{rad}^* - yF_{oop}^* \end{pmatrix} \tag{42}$$

The $x-$component of the aerodynamic moment $\boldsymbol{M}$ is the contribution to aerodynamic torque. Then, the aerodynamic power

of the rotor is the integrated contribution of $M_x$ of all $N_B$ blades at rotational speed of $\Omega$:

$$P = N_B \Omega \int_0^{R_{\text{tot}}} yF_{ip}^* \, \mathrm{d}r \tag{43}$$

The aerodynamic thrust of the rotor is the total contribution of the out-of-plane force of all $N_B$ blades:

$$T = N_B \int_0^{R_{\text{tot}}} F_{oop}^* \, \mathrm{d}r \tag{44}$$

The aerodynamic power and thrust of the rotor with baseline straight blades without cone as well as the rotors with dihedral

blades without cone are calculated. It is difficult to directly draw conclusions from the results of power and thrust. To better illustrate and compare the integral effects of the rotor dihedral represented by the aerodynamic power and thrust predicted using different methods, the relative difference of the aerodynamic power and thrust with respect to the baseline rotor from each method are calculated and are summarized in Table 2 and 3.

For the aerodynamic power, the relative change predicted by LL is underestimated compared to the prediction by CFD but

the results are showing similar trends. One of the reasons could be the use of the 2-D airfoil data in the lifting-line method. The ordinary BEM method predicts almost no influence of dihedral on power, only except for W-1 that predicts the correct direction that the power increases but underestimates the magnitude. The BEM-$u_r$ method predicts the same direction as CFD and LL that the power of the dihedral rotors are increased compared to the baseline. However, the magnitude of the increment





is overestimated by a factor of approximately two. Comparing to the other two BEM methods, the relative increment of power predicted by the proposed BEM-VC method is in better agreement with the predictions by CFD and LL.

For the aerodynamic thrust, the magnitude of the relative decrement is overestimated by approximately 20% by LL comparing to CFD. The BEM method predicts the correct trend that the thrust decreases but the magnitude is underestimated compared to LL and CFD. The BEM-$u_r$ method predicts a small increase of aerodynamic thrust instead of a decrease of thrust as predicted by the higher-fidelity models. The relative change of the thrust predicted by the proposed BEM-VC method is very similar to the results predicted by LL, and the relative difference of the predicted relative change is less than 20%.

**Table 2.** The aerodynamic power (in kW) of the baseline straight blade and the relative difference in aerodynamic power (in %) of the different dihedral blades with respect to the baseline blades calculated using different aerodynamic models. The operational condition is with uniform wind speed of $8\,\mathrm{m\,s^{-1}}$, rotational speed $0.855\,\mathrm{rad\,s^{-1}}$ and zero cone angle.

| Name | CFD | LL | BEM | BEM-$u_r$ | BEM-VC |
|------|------|------|------|------|------|
| Baseline | 4358.6 | 4501.6 | 4450.8 | 4450.5 | 4451.8 |
| W-1 | 3.13% | 2.05% | 1.01% | 6.24% | 1.48% |
| W-2 | 2.89% | 1.91% | 0.31% | 6.04% | 1.32% |
| W-3 | 1.73% | 0.92% | 0.24% | 3.34% | 0.71% |
| W-4 | 1.57% | 0.71% | -0.02% | 2.91% | 0.66% |

**Table 3.** The aerodynamic thrust (in kN) of the baseline straight blade and the relative difference in aerodynamic thrust (in %) of the different dihedral blades with respect to the baseline blades calculated using different aerodynamic models. The operational condition is with uniform wind speed of $8\,\mathrm{m\,s^{-1}}$, rotational speed $0.855\,\mathrm{rad\,s^{-1}}$ and zero cone angle.

| Name | CFD | LL | BEM | BEM-$u_r$ | BEM-VC |
|------|------|------|------|------|------|
| Baseline | 1070.7 | 1086.9 | 1084.0 | 1084.2 | 1083.9 |
| W-1 | -1.18% | -1.46% | -0.95% | 0.29% | -1.63% |
| W-2 | -1.34% | -1.53% | -1.02% | 0.22% | -1.63% |
| W-3 | -0.58% | -0.75% | -0.40% | 0.28% | -0.70% |
| W-4 | -0.60% | -0.76% | -0.42% | 0.20% | -0.63% |

In summary, the proposed BEM-VC model is in better agreement with higher-fidelity models when predicting the integrated aerodynamic power and thrust of the dihedral rotor, compared to the ordinary BEM method. The BEM method with radial induction correction (BEM-$u_r$) is predicting worse compared to the BEM method for both aerodynamic power and thrust.





## 8 Conclusions and future work

A new computationally efficient method for the aerodynamic load calculation of non-planar rotors is described. The method is based on the vortex cylinder model, and can be used in two ways: either as a correction to the currently widely used blade element momentum (BEM) method, or used as the main model, replacing the BEM method in the engineering modelling

complex. For the uniform inflow that is perpendicular to the rotor plane, the influence of the blade out-of-plane shapes on the distributed aerodynamic loads, measured by the difference of the loads between the non-planar rotor and the planar rotor, is shown to be in good agreement with higher-fidelity models. The predicted distributed and integrated aerodynamic loads are also in better agreement with higher-fidelity models, comparing to the BEM method and also the BEM method with radial induction correction. While the present work focused on stiff geometries, the developed framework would be able to handle

out-of-plane deflections during aeroelastic simulations accounting for blade elasticity, without any loss of generality. The new model is approximately as numerically efficient as ordinary BEM-based models, which makes it favourable for aero-servo-elastic simulation as well as design optimization of horizontal-axis wind turbines whose blades have out-of-plane shapes. Therefore, the authors recommend the use of the proposed model as a correction to the existing BEM codes.

There are several future works needed to further improve the model. Firstly, it is favourable to have modifications to the

existing Prandtl's tip-loss correction. For example, it is possible to use the distance between the tip vortex and the calculation point when calculating the correction for a non-planar rotor, instead of using the radial distance as currently implemented in the model. Secondly, it is beneficial to further develop the model for the application of blades with both in-plane and out-of-plane shapes. One possible track of the development is to couple the vortex cylinder model and the near-wake model, which approximately models the near-wake as helical trailed vorticities and is currently coupled with a far-wake BEM method.

Thirdly, it is interesting to use both the BEM method and the proposed method for the aerodynamic or aeroelastic design of a non-planar rotor under some constraints. Higher-fidelity models, such as CFD or lifting-line method can be used for the benchmark of different designs, as done for the present work. Finally, further development of the model focusing on analytical gradients is favourable for the application in a gradient-based wind turbine design optimization framework.

*Author contributions.*   AL conducted the study as part of his PhD research. The idea of the proposed model originates from MG. The proposed model is jointly developed by MG, AL and GRP. The similarities of the superposition of the vortex cylinder model of non-planar rotor and planar rotor are described by AL and MG. The application of the tip-loss correction as well as high-thrust correction for the non-planar rotor is described by AL and MG. The implementation of the proposed model in HAWC2 code and the computations using the HAWC2 code are performed by AL with contribution from GRP. The CFD method is introduced by SGH and the CFD results are computed by SGH. The

post-processing of the CFD results is performed by SGH with contribution from AL. The lifting-line results are computed by AL and the post-processing is performed by AL. All authors jointly draw the conclusions of the work and contribute to writing this work.



*Competing interests.* DTU Wind Energy develops and distributes HAWC2 on commercial and academic terms.

*Acknowledgements.* The authors would like to thank our colleague Néstor Ramos García in DTU Wind Energy for the suggestions in the lifting-line simulation using MIRAS, a vortex code mainly developed by him. This work has been supported by the Smart Tip project, funded
5 by Innovation Fund Denmark (J.nr. 7046-00023B).



## Appendix A: Nomenclature

$a$    axial induction

$a'$    tangential induction

$C_L$    lift coefficient

$C_D$    drag coefficient

$C_m$    moment coefficient

$C_Q$    torque coefficient

$C_T$    thrust coefficient

$C_{T,\mathrm{eff}}$    effective thrust coefficient

$C_{T,\mathrm{KJ}}$    Kutta-Joukowski thrust coefficient

$C_{T,\mathrm{rot}}$    thrust coefficient due to wake rotation

$C_{T,av}$    averaged coefficient for the radial induction function

$\Delta d$    dihedral magnitude

$\boldsymbol{f}$    lift force vector on the blade with the definition of force per unit length of curved blade length

$\boldsymbol{f^*}$    lift force vector on the blade with the definition of force per unit radius

$\boldsymbol{F^*}$    aerodynamic force vector on the blade with the definition of force per unit radius

$F$    tip-loss factor

$h$    helical pitch

$k$    factor for the calculation of the elliptic integral

$k_1, k_2, k_3$    factors for the relationship between axial induction and thrust coefficient

$k_s$    normalized sectional circulation of the vortex cylinder

$\boldsymbol{M}$    the aerodynamic moment vector

$N_B$    number of blades

$P$    aerodynamic power of the rotor

$r$    radius of the calculation point

$R$    radius of the vortex cylinder

$R_{\mathrm{tot}}$    radius of the rotor

$T$    aerodynamic thrust of the rotor

$u_a$    axial induced velocity

$u_t$    tangential induced velocity

$u_r$    radial induced velocity

$\Delta u_a$    the correction to the axial induced velocity

$\Delta u_r$    the correction to the radial induced velocity

$U_0$    wind speed





$V_{rel}$     relative velocity

$x$     axial position of the calculation point with respect to the vortex cylinder

**Greek letters**

$\Gamma$     bound vorticity strength of the vortex cylinder, equal to the bound vorticity strength of all blades

$\Gamma^B$     blade bound vorticity strength

$\Delta\Gamma$     trailed vorticity strength of the vortex cylinder

$\gamma_t$     tangential vorticity strength of the vortex cylinder

$\Lambda_{tip}$     tip dihedral angle

$\varphi$     inflow angle

$\rho$     density of air

$\theta_c$     cone angle

$\kappa$     dihedral angle

$\Omega$     rotor speed

$\lambda_r$     speed ratio at radius $r$

**Subscripts**

$a$     in the axial direction

$t$     in the tangential direction

$r$     in the radial direction

$i$     at blade section $i$

$B$     the value at the blade

$\infty$     the annulus averaged value

eff     effective value

tot     the total value

MT     from the momentum theory

BEM     from the BEM method

VC     from the vortex cylinder model

**Superscripts**

$B$     the value at the blade

np     the non-planar rotor

pl     the planar rotor

$QS$     quasi-steady

$C$     circulatory part





## Appendix B: Results of the distributed load

### B1  Zero cone angle

The difference of the loads of the dihedral blades (W-2 to W-4) with zero cone compared to the baseline straight blade without cone.

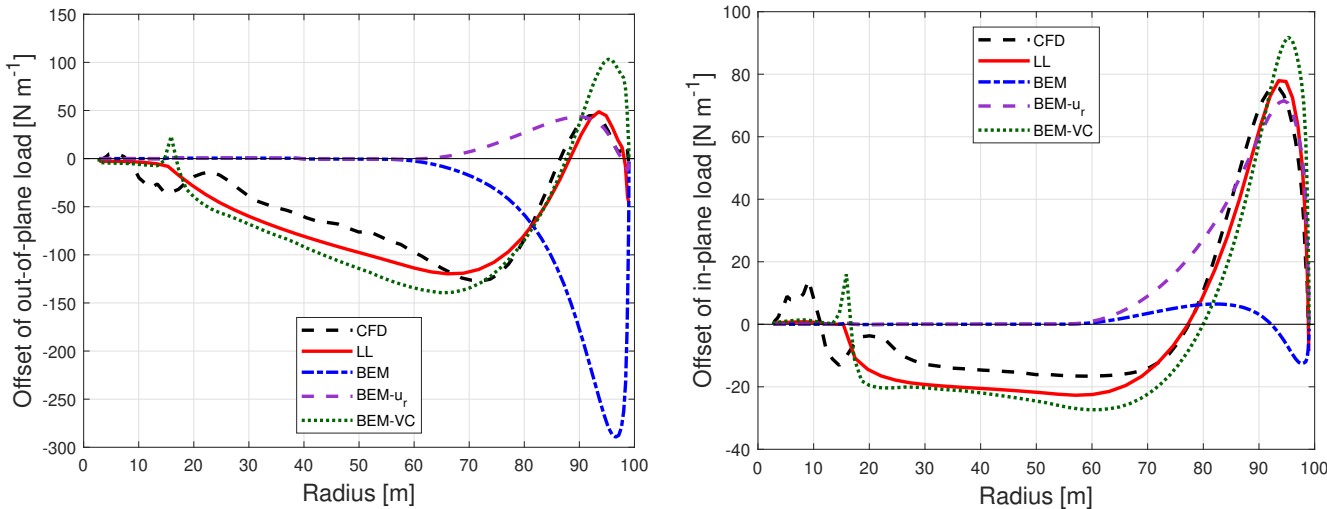

**Figure B1.** Comparison of the difference of the out-of-plane load (left) and in-plane load (right) of the dihedral blade W-2 compared to the baseline blade calculated from CFD, lifting-line method (LL), BEM method, BEM with radial induction (BEM-$u_r$) and the proposed BEM-VC method.

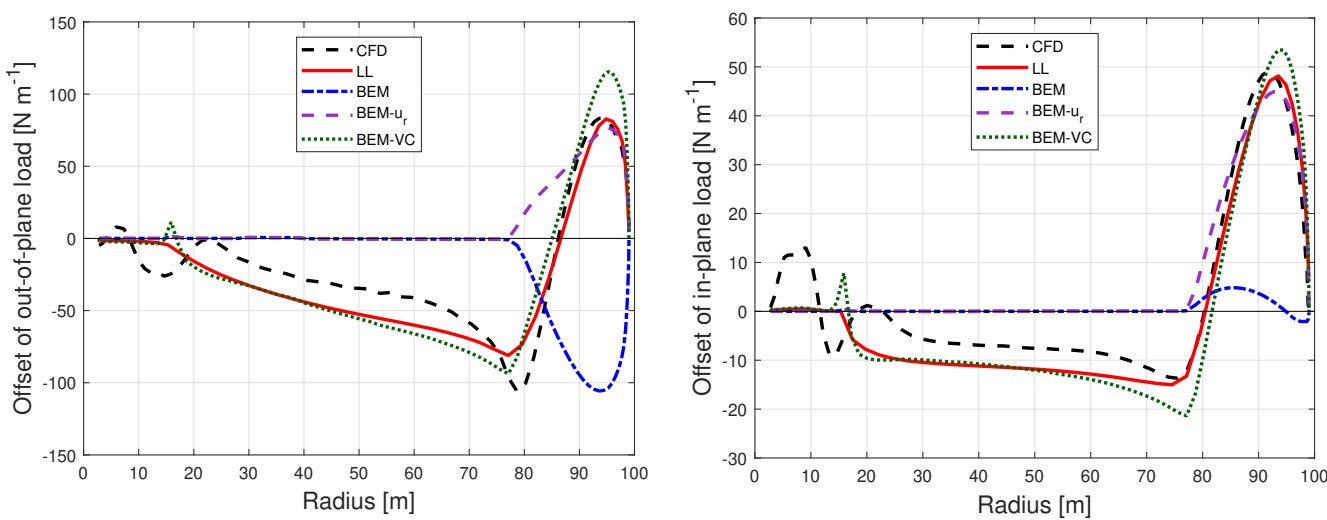

**Figure B2.** Comparison of the difference of the out-of-plane load (left) and in-plane load (right) of the dihedral blade W-3 compared to the baseline blade calculated from different aerodynamic models.



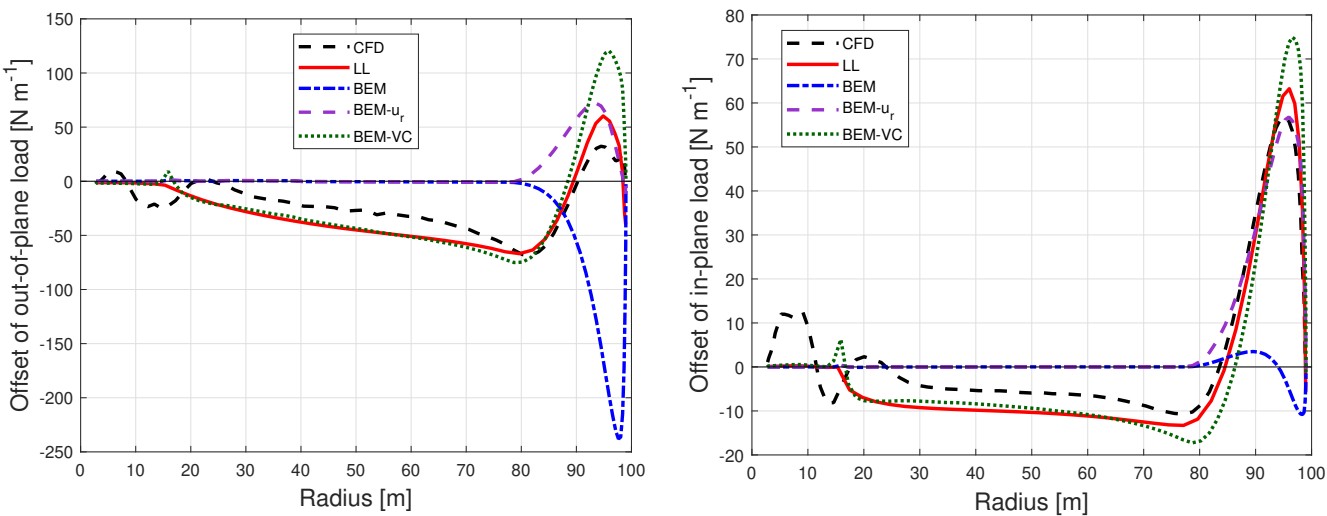

**Figure B3.** Comparison of the difference of the out-of-plane load (left) and in-plane load (right) of the dihedral blade W-4 compared to the baseline blade calculated from different aerodynamic models.

## B2 Upwind cone

The difference of the loads of the dihedral blades (W-2 to W-4) with $15°$ cone upwind compared to the baseline straight blade with the same upwind cone.

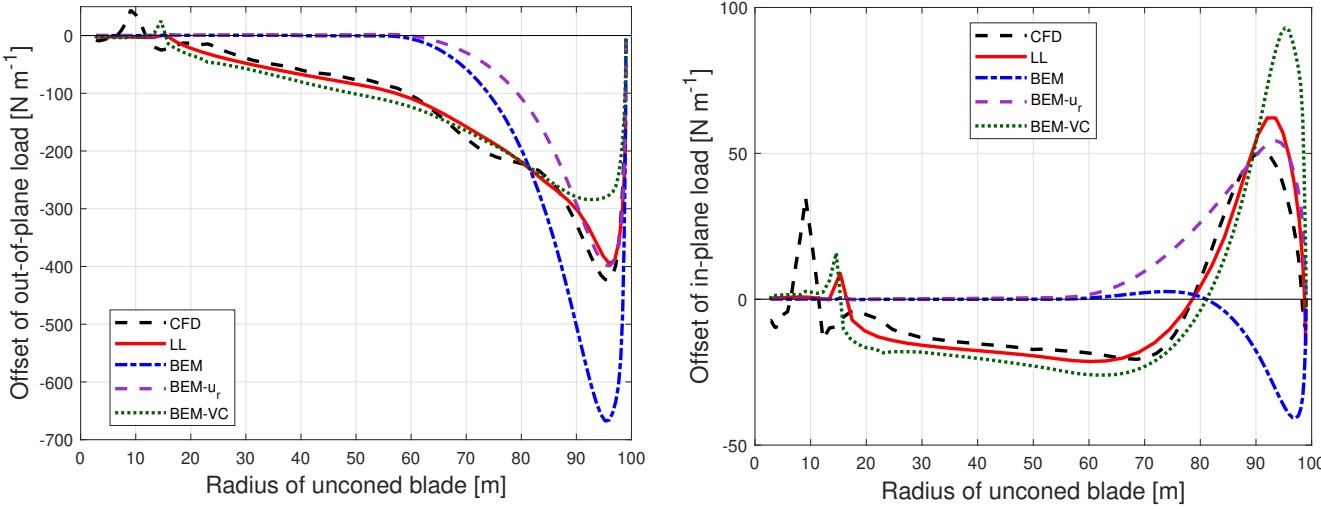

**Figure B4.** Comparison of the difference of the out-of-plane load (left) and in-plane load (right) of the dihedral blade W-2 with $15°$ of cone upwind compared to the baseline blade with the same cone calculated from CFD, lifting-line method (LL), BEM method, BEM with radial induction (BEM-$u_r$) and the proposed BEM-VC method.



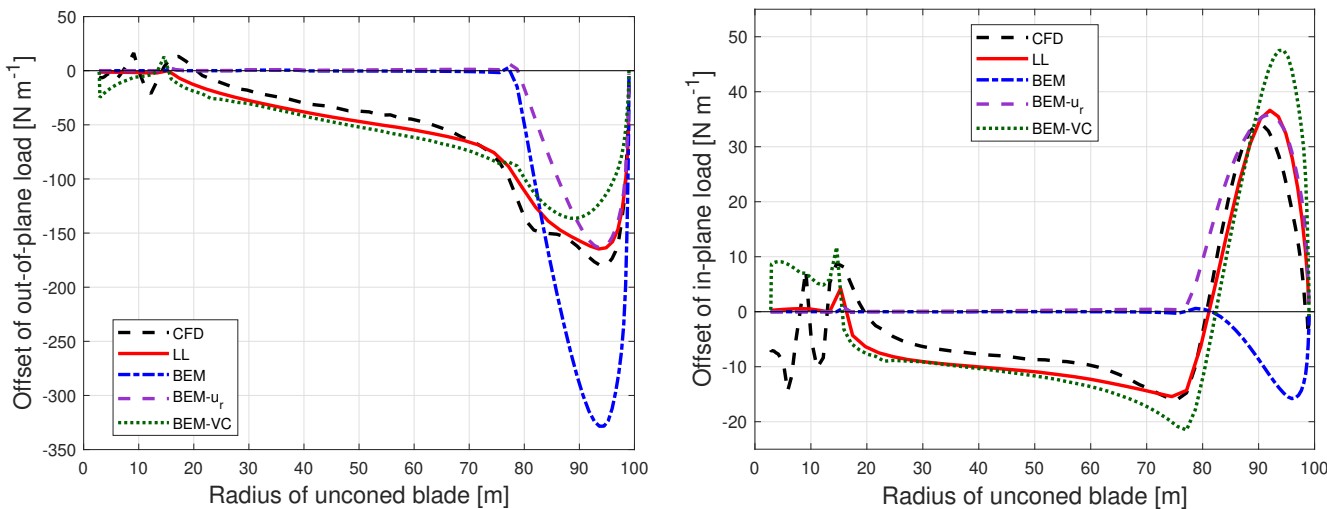

**Figure B5.** Comparison of the difference of the out-of-plane load (left) and in-plane load (right) of the dihedral blade W-3 with 15° of cone upwind compared to the baseline blade with the same cone calculated from different aerodynamic models.

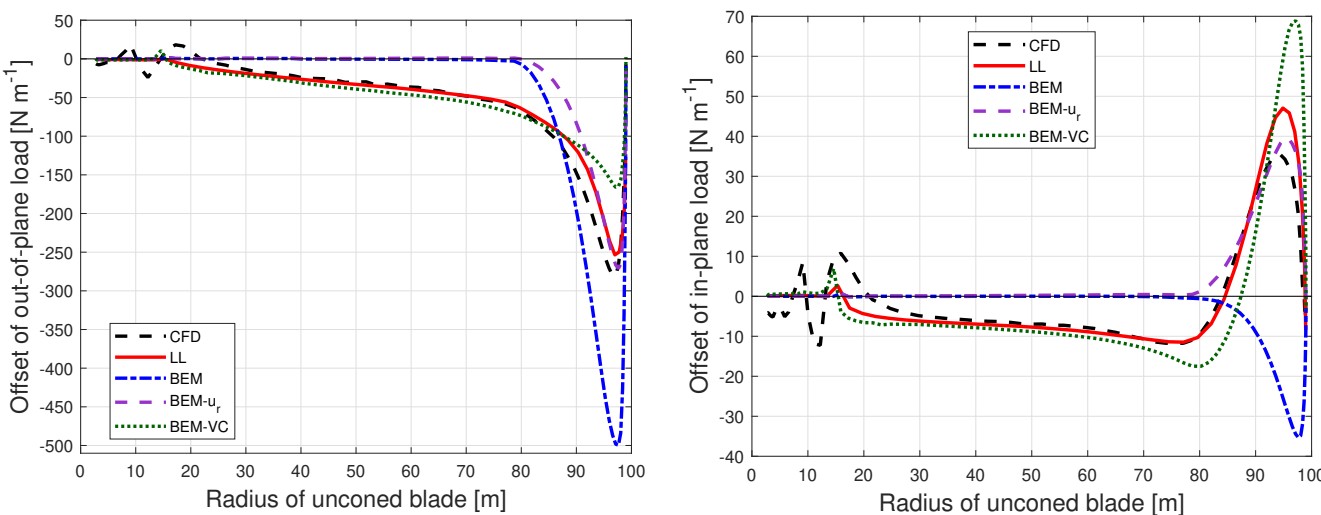

**Figure B6.** Comparison of the difference of the out-of-plane load (left) and in-plane load (right) of the dihedral blade W-4 with 15° of cone upwind compared to the baseline blade with the same cone calculated from different aerodynamic models.



## B3 Downwind cone

The difference of the loads of the dihedral blades (W-2 to W-4) with 15° cone downwind compared to the baseline straight blade with the same downwind cone.

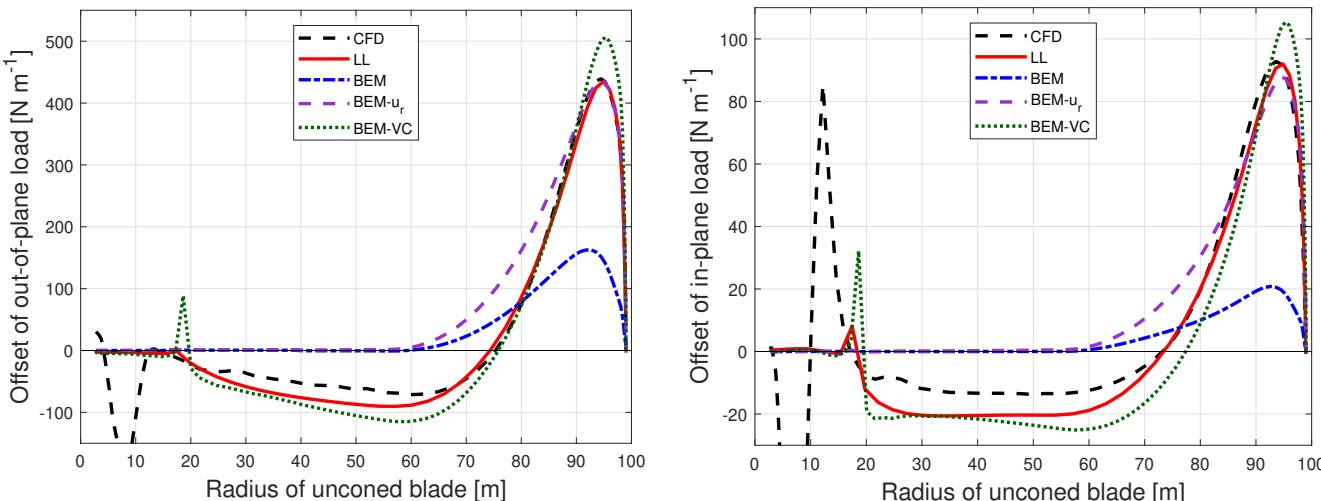

**Figure B7.** Comparison of the difference of the out-of-plane load (left) and in-plane load (right) of the dihedral blade W-2 with 15° of cone downwind compared to the baseline blade with the same cone calculated from CFD, lifting-line method (LL), BEM method, BEM with radial induction (BEM-$u_r$) and the proposed BEM-VC method.

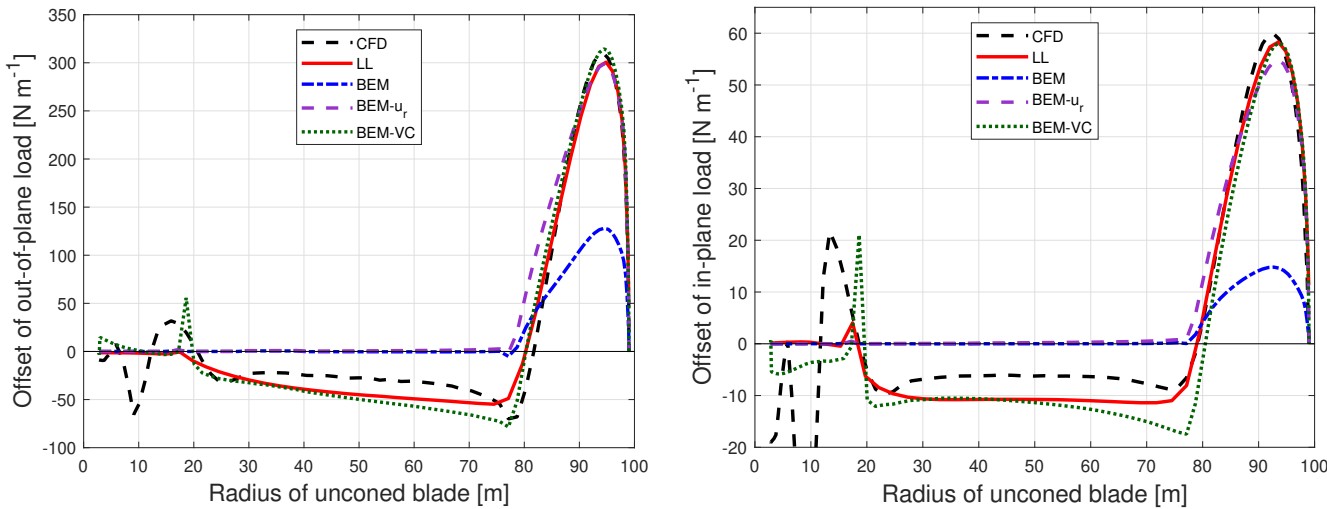

**Figure B8.** Comparison of the difference of the out-of-plane load (left) and in-plane load (right) of the dihedral blade W-3 with 15° of cone downwind compared to the baseline blade with the same cone calculated from different aerodynamic models.





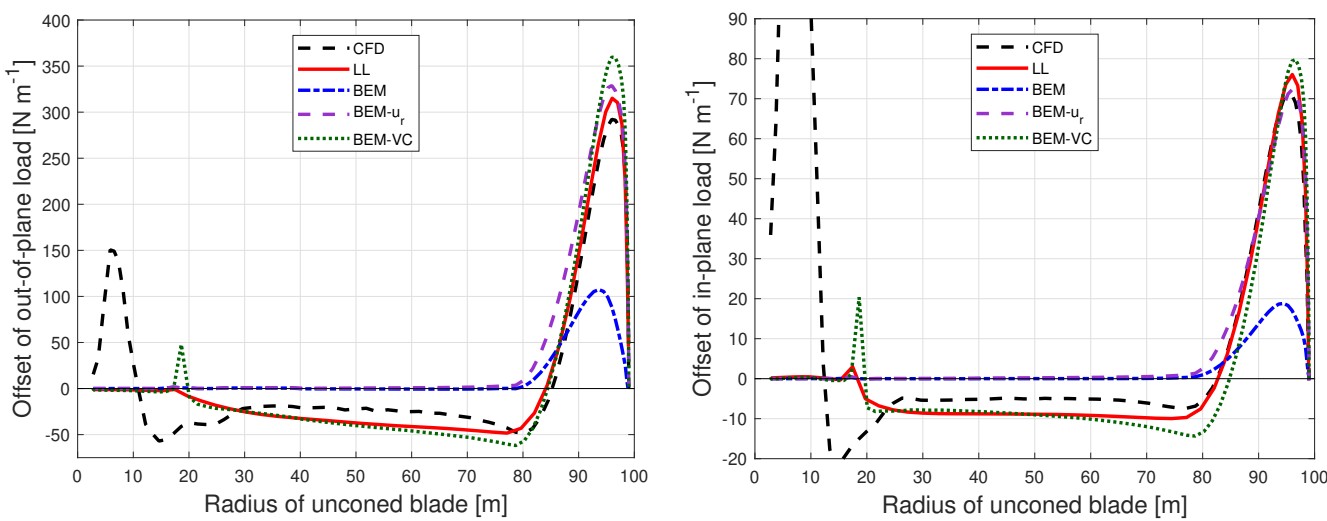

**Figure B9.** Comparison of the difference of the out-of-plane load (left) and in-plane load (right) of the dihedral blade W-4 with 15° of cone downwind compared to the baseline blade with the same cone calculated from different aerodynamic models.





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
