# Peer review of "A computationally efficient engineering aerodynamic model for non-planar wind turbine rotors"

_Wind Energy Science, 2021_

## Author Comment (AC1)

November 25, 2021

Dear Reviewers

First of all, we would like to thank you for the constructive and detailed comments on our article. Please find below our responses (in black) to your comments (in blue). In addition to changes suggested by the reviewers, we updated the definition of the loads from out-of-plane, in-plane and radial loads to axial, tangential and radial loads, which is more precise. The other content in the figures are not changed. At the end of this letter, we have included a marked-up version showing all changes in the manuscript.

Yours sincerely

A. Li, M. Gaunaa, G. R. Pirrung and S. G. Horcas

Technical University of Denmark          Frederiksborgvej 399          angl@dtu.dk
**Department of Wind Energy**          Building 125          www.dtu.dk
          DK-4000 Roskilde

[Figure]

**Comments to Reviewer #1**

The authors developed a new methodology for the simulation of Horizontal-Axis Wind Turbines (HAWTs) with dihedral blades, i.e. developing out of the rotation plane of the rotor, and not negligible coning. This approach combines the standard Blade Element Momentum (BEM) with a new correction based on the Vortex Cylinder (VC) model previously developed by the authors. The proposed method was tested on a selected large size turbine for different blade out-of-plane bending and rotor cone angles and the obtained predictions were compared with those of the BEM and two high-order methods, i.e. Lifting Line Theory (LLT) and blade-resolved CFD, in terms of blade spanwise load distributions. A promising accuracy improvement is observed with the new method compared to traditional BEM.

The reviewer believes that the topic and the activity is very interesting, innovative and worthy of investigation. The adopted methodology is rigorous and clearly detailed throughout the whole paper, which is very well presented. Some specific considerations:

Section 3: it would be helpful for the reader to add a schematic representation of the vortex cylinder method, in which the different components of the formulation – root vortex, vortex cylinder and bound vortex disc – are clearly highlighted in the turbine rotor frame

Based on the aforementioned comments, the publication of the paper in the present form is strongly recommended.

Thank you for your comments. We included an illustration of the different components of the vortex cylinder model in Fig. 2.

[Figure]

**Comments to Reviewer #2**

A well written and interesting paper about a fast rotor aerodynamic performance prediction method to simulate out of plane geometries. This is believed to contain a nice contribution to the field.

General comments:

Although the current paper already includes a significant amount of work, it could be recommended to study more operating points along the power curve (e.g. instead of focusing on different cone and dihedral shapes) to also investigate the effectiveness for these different conditions? If the application of this method truly lies in design and optimization it would be good to base the conclusions on more than a single operating point.

Thank you for the comment. We agree it is important to show more operating conditions. We included a new section (Sect. 7.4) that is focused on a single dihedral blade geometry (W-1) operating at three different operational conditions that have lower loading, compared to the high loading cases that are previously shown. The three cases correspond to rotor thrust coefficients of approximately 0.4, 0.2 and 0.1, respectively.

Along this line of recommendations also lies the application of this method to unsteady aerodynamic or aero-elastic phenomena, e.g. quantifying its impact on aerodynamic damping in turbulent inflow.

The aerodynamic damping on the blade and including the current method into an aeroelastic simulation involve unsteady aerodynamic effects. We think that future work on the unsteady effect of non-planar rotors is interesting to be investigated. We added sentences in Sect. 8 to highlight this future work:

*Thirdly, it would be interesting to investigate the unsteady effects of the non-planar rotor, such as aerodynamic damping and dynamic inflow effect.*

For the prediction of the aerodynamic damping in rotation and under turbulent inflow, we consider there will be almost no difference between the proposed vortex cylinder model and the momentum theory. This is because the fast response is represented by the 2-D airfoil aerodynamic (dynamic stall model) and the slow response is represented by the dynamic inflow effect. For the aeroelastic simulations, the existing dynamic inflow model assumes a planar rotor. However, the vortex cylinder model can be further modified to account for the dynamic inflow characteristics of non-planar rotors.

It is mentioned that the computational effort is similar to a BEM method. Can this be further quantified also in relation to the LL and CFD simulations?

We included a new section (Sect. 7.5) that is dedicated to describing the computational effort of different aerodynamic models that are used in the present work. The high performance computing (HPC) cluster setup and the corresponding wall clock time of the RANS Navier-Stokes solver, as well as the lifting-line method, are presented. The CPU time for the BEM method and the proposed

vortex cylinder method is also described.

*2 p2 line 11/12 Can the authors supply a reference for this claim?*

We included an reference by Madsen and Rasmussen [1].

*3 p6 line 17: The application of the vortex cylinder is mentioned for modeling yawed flow. Can the authors comment on the useability of the current implementation focusing on out of plane shapes for the modeling of yawed flow?*

We believe that the method can be extended for the modelling of a non-planar rotor under yawed flow, but future work is needed. The work would be based on the present work and the previous work by Branlard and Gaunaa [2]. We included a sentence in Sect. 8 to highlight this aspect as future work:

*Fourthly, it is beneficial to further develop the vortex cylinder model for the application of non-planar rotors in yawed flow.*

*7.1 p24 What kind of Cdax, a are we at? AOA? Some idea would be nice.*

We included the sentences that describes the operational condition of the baseline straight blade case:

*At this operational condition, the thrust coefficient of the unconed rotor with baseline straight blades is 0.90 and the rotor power coefficient is 0.46, as predicted by the BEM method. At radius of 70 m, the angle of attack predicted by the BEM method is 5.76°.*

*7.2.1 p25 Fig. 9. The authors rightly mention it would be difficult to draw conclusions based on the absolute loads plots and therefore proceed with a plot relative to the baseline case. Can the authors perhaps comment on the fact that the difference between the LL and BEM simulation for the baseline case is of a similar magnitude compared to the effect of dihedral?*

Despite the difference between the LL method and the BEM/BEM-VC methods for the baseline case, the BEM-VC method is favourable for the application in a multi-fidelity optimization framework because it can model the influence of blade dihedral. We included the following paragraph in Sect. 7.2.2 to describe this:

*It should be mentioned that the difference between the higher-fidelity models (CFD and LL) and the BEM-VC method for the baseline straight blade is of similar magnitude as the influence due to blade dihedral. However, since the model predicts the sensitivity of changes in dihedral relatively well, it is favourable to be used for parameter studies and to be eventually integrated in a multi-fidelity aerodynamic optimization framework. For example, in order to design a rotor with dihedral blades, higher-fidelity models could be used for the initial design of a straight blade. Then, the proposed vortex cylinder model could be used to explore the sensitivity of different dihedral*

[Figure]

*parameters on the aerodynamic loads, with a relatively low computational effort.*

7.2.2 p27 Line 9/10 Can the authors comment on the physics behind the observation of the impact of dihedral inboard of the start of the dihedral?

We included the following sentences in Sect. 7.2.2 to describe the difference of the effects of the blade dihedral and the blade sweep using vortex theory:

*This could be explained by the out-of-plane blade shape moving the starting position of the trailed vortex system in the axial direction. That, in turn, could effectively move the inner parts of the rotor further into or out of the induction field created by the vortex sheets trailed from the outer sections. By contrast, the influence of blade sweep on the trailed vortex is only on the azimuthal starting position of the trailed vortex. If we consider the trailed vortex as a frozen helical wake, sweeping the blade would only result in an azimuthal twisting of the helical wake. Therefore, the shape of the vortex wake would almost remain unchanged, so that the influence on the induction would not be global.*

p32 Table 2 and 3 are perhaps better visualized in a line or bar plot??

We agree that the bar plot visualizes the results better compared to the table. We have replaced two tables with two bar plots in Sect. 7.3. The aerodynamic power and thrust of the rotor with baseline straight blades in the two tables (previously Table 2 and 3) are summarized into a new table (Table 2).

[Figure]

**Bibliography**

[revised manuscript text omitted]

$$u_{t,\text{MT}} = u_{t,\text{VC}}^{\text{pl}} \tag{30b}$$

$$u_{r,\text{MT}} \equiv 0 \tag{30c}$$

**3.5.2 Non-planar rotor**

For a non-planar rotor, the inductions from the momentum theory with wake rotation effect included are equivalent to the inductions from the vortex cylinder model for the corresponding planar rotor and excluding the radial induced velocity. This means the momentum theory equivalently assumes the rotor is planar when calculating inductions. Then, the vortex cylinder model for the non-planar rotor is equivalent to the momentum theory with the following corrections: For the annulus axial induced velocity, the correction is the difference of the results of the non-planar rotor and the corresponding planar rotor from the vortex cylinder model. The tangential induced velocity from both methods are the same, as described in Sect. 3.4.3. The radial induction is not available from the momentum theory, so the correction should be the complete radial induction of the non-planar rotor from the vortex cylinder model.

These relationships between the momentum theory and the vortex cylinder model for the non-planar rotor are summarized in the equations as follows, where the superscript of np represents the non-planar rotor.

$$u_{a,\text{VC}}^{\text{np}} = u_{a,\text{MT}} + \Delta u_a \tag{31a}$$

$$u_{t,\text{VC}}^{\text{np}} = u_{t,\text{MT}} \tag{31b}$$

[revised manuscript text omitted]